# RecDreamer: Consistent Text-to-3D Generation via Uniform Score Distillation

**Chenxi Zheng[1], Yihong Lin[1], Bangzhen Liu[1], Xuemiao Xu[1][†], Yongwei Nie[1][†], Shengfeng He[2]**
[1]South China University of Technology, [2]Singapore Management University
{chansey0529, amcsyihonglin, liubz.scut}@gmail.com, {xuemx, nieyongwei}@scut.edu.cn,
shengfenghe@smu.edu.sg

## Abstract

Current text-to-3D generation methods based on score distillation often suffer from geometric inconsistencies, leading to repeated patterns across different poses of 3D assets. This issue, known as the Multi-Face Janus problem, arises because existing methods struggle to maintain consistency across varying poses and are biased toward a canonical pose. While recent work has improved pose control and approximation, these efforts are still limited by this inherent bias, which skews the guidance during generation. To address this, we propose a solution called RecDreamer, which reshapes the underlying data distribution to achieve a more consistent pose representation. The core idea behind our method is to rectify the prior distribution, ensuring that pose variation is uniformly distributed rather than biased toward a canonical form. By modifying the prescribed distribution through an auxiliary function, we can reconstruct the density of the distribution to ensure compliance with specific marginal constraints. In particular, we ensure that the marginal distribution of poses follows a uniform distribution, thereby eliminating the biases introduced by the prior knowledge. We incorporate this rectified data distribution into existing score distillation algorithms, a process we refer to as uniform score distillation. To efficiently compute the posterior distribution required for the auxiliary function, RecDreamer introduces a training-free classifier that estimates pose categories in a plug-and-play manner. Additionally, we utilize various approximation techniques for noisy states, significantly improving system performance. Our experimental results demonstrate that RecDreamer effectively mitigates the Multi-Face Janus problem, leading to more consistent 3D asset generation across different poses.

## 1 Introduction

Text-to-3D generation has become a transformative technology with broad applications, enabling the creation of 3D models from natural language descriptions. By lowering the technical barriers, it allows non-experts to generate intricate 3D objects without specialized tools or expertise. This advancement significantly enhances productivity in fields such as gaming, virtual reality (VR), and augmented reality (AR), where manual 3D model creation is often labor-intensive. Current methods (Wang et al., 2024b; Chen et al., 2023; Lin et al., 2023) rely on score distillation techniques (Poole et al., 2022; Wang et al., 2023a; Graikos et al., 2022) to leverage text-to-image priors from diffusion models, generating high-quality 3D assets with remarkable visual fidelity, precise alignment to text descriptions, and strong conceptual integrity.

However, despite these advances, generated 3D assets frequently suffer from geometric inconsistencies, particularly in the form of repeated patterns or textures across different camera poses, a problem known as the *Multi-Face Janus* issue (see Fig. 1(b)). This arises from biases in the underlying data distribution (see Fig. 1(a)), which current methods fail to fully address. Efforts to tackle this issue, such as modifying directional text descriptions through gradient-based adjustments (Hong et al., 2023; Armandpour et al., 2023), have yielded limited success, often introducing unwanted artifacts or irrelevant patterns. Other approaches (Huang et al., 2024; Wang et al., 2024a) attempt to impose

---

[†]Corresponding authors. Code: `https://github.com/chansey0529/RecDreamer`.

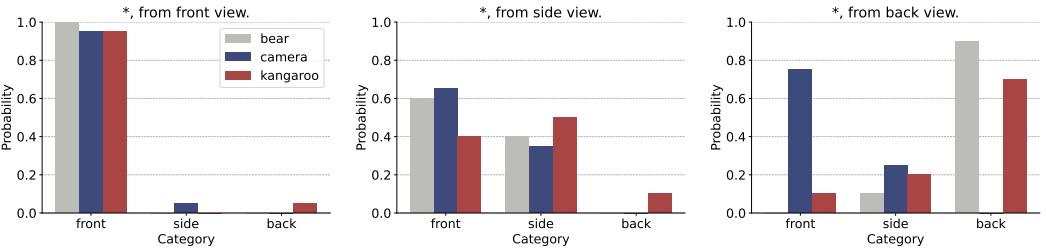

(a) Statistics for pose classification in images generated by diffusion models. The plots show that the overall distribution is biased toward front-facing poses, even when directional text prompts are used to specify other poses.

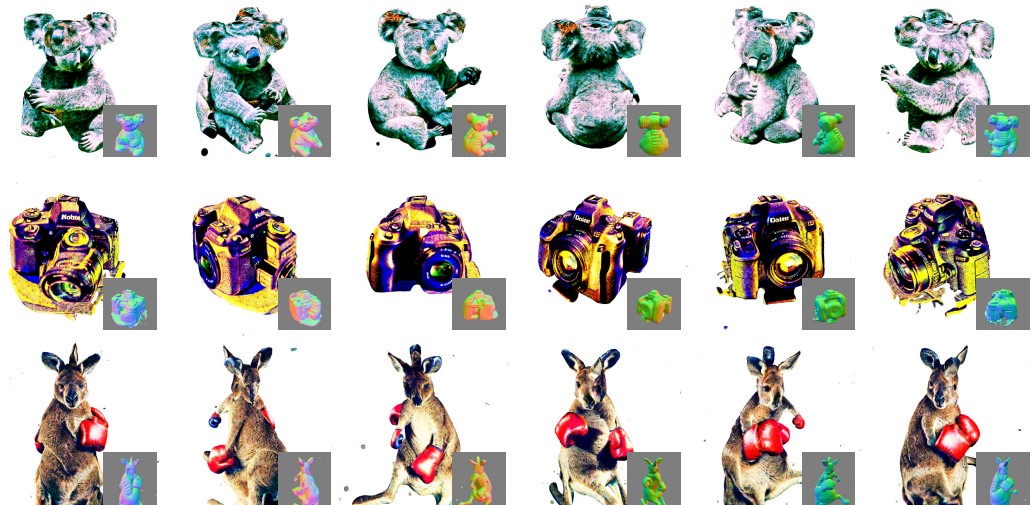

(b) Score distillation from a biased distribution leads to the Multi-Face Janus problem. The generated 3D assets tend to overemphasize frontal features to align with the prior distribution, resulting in repeated patterns.

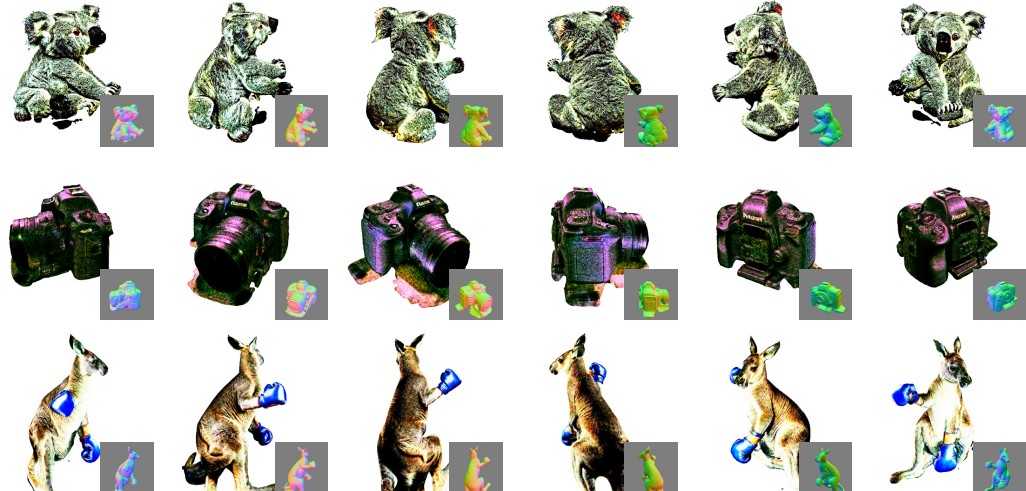

(c) Score distillation using a uniform distribution with our RecDreamer. The process relies on a rectified prior distribution that incorporates guidance from various poses, effectively alleviating geometric inconsistencies.

Figure 1: The Multi-Face Janus problem arises from an imbalance in the pose distribution of pre-trained models, which tend to generate predominantly frontal images. This bias results in excessive faces appearing in the generated 3D assets. RecDreamer addresses this issue by producing a distribution with a uniform pose marginal, enabling more diverse pose generation and mitigating the Multi-Face Janus problem.

constraints on the rendered 3D assets, but they still fall short of resolving the core bias present in text-to-image distributions.

To address this, we propose *RecDreamer*, a novel solution designed to eliminate the biases in pre-trained models by modifying the underlying data distribution. The rationale behind our approach is to reconstruct the original data distribution so that the marginal distribution of the pose becomes uniform, thus removing the bias toward a canonical pose. We achieve this by introducing a weighting function that reweights the density of the original distribution, ensuring it meets specific marginal constraints. Specifically, we derive a rectified distribution where the pose component in the joint distribution follows a uniform distribution across all possible poses.

This rectified distribution is then incorporated into the score distillation framework (Wang et al., 2024b). The use of reverse Kullback-Leibler divergence (Kullback & Leibler, 1951) in score distillation allows the integration of the modified distribution without altering the overall sampling process or gradient derivation. As a result, we develop a process known as uniform score distillation (USD), which aligns the target distribution with a uniform distribution, effectively improving pose consistency in the generated 3D assets.

To compute the auxiliary function necessary for rectifying the distribution, RecDreamer introduces a training-free classifier that estimates pose categories by discretizing the continuous pose space. This classifier predicts pose based on orientation score and texture similarity, leveraging a pretrained feature extractor without the need for additional fine-tuning. Furthermore, we dynamically handle noisy image estimates, ensuring robust pose estimation and reliable performance even in the optimization process.

Experiments demonstrate the effectiveness of our method in alleviating the Multi-Face Janus problem and improving geometric consistency, while maintaining rendering quality comparable to baseline methods, as shown in Fig. 1(c). We also conduct additional experiments on 2D images and a toy dataset to further validate our algorithm. Additionally, we showcase further applications of the pose classifier.

## 2 BACKGROUND

In this section, we provide a brief overview of diffusion models, conditional guidance techniques, and text-to-3D generation using score distillation. We follow the notation conventions introduced in VSD (Wang et al., 2024b).

### 2.1 DIFFUSION MODELS

Diffusion models (Sohl-Dickstein et al., 2015; Ho et al., 2020; Song et al., 2020) are latent variable models that simulate a diffusion process to model the data distribution $x_0 \sim q_0(x_0)$. These models consist of a forward process $q$, which progressively adds Gaussian noise to the data, and a reverse process $p$, which denoises the data to recover the original distribution.

In the forward process, noise is iteratively added through transitions $q_t(x_t|x_{t-1})$. This allows the posterior distribution $q_t(x_t|x_0) = \mathcal{N}(\alpha_t x_0, \sigma_t^2 I)$ to be computed, where $\alpha_t$ and $\sigma_t$ are time-dependent hyperparameters. The marginal distribution $q_t(x_t)$ is derived by integrating over the data distribution:

$$q_t(x_t) = \int q_t(x_t|x_0)q_0(x_0) \, dx_0. \tag{1}$$

The reverse process begins with a standard Gaussian distribution, $p_T(x_T) = \mathcal{N}(0, I)$, and removes the noise through transitions $p_t(x_t|x_{t+1}) = q_t(x_t|x_{t+1}, x_0 = \hat{x}_0)$, where $\hat{x}_0$ is an estimate of the clean data. Instead of predicting $\hat{x}_0$ directly, Ho et al. (2020) proposed optimizing a noise estimator $\epsilon_\phi(x_t, t)$ by minimizing the following loss function:

$$\mathcal{L}_{\text{Diff}}(\phi) = \mathbb{E}_{x_0 \sim q_0(x_0), t \sim \mathcal{U}[0,T], \epsilon \sim \mathcal{N}(0,I)} \left[ \omega(t) \| \epsilon_\phi(\alpha_t x_0 + \sigma_t \epsilon) - \epsilon \|_2^2 \right], \tag{2}$$

where $\omega(t)$ is a time-dependent weighting function. The noise predictor $\epsilon_\phi(x_t, t)$ can be viewed as a linear transformation of the score function: $\epsilon_\phi(x_t, t) = -\sigma_t s_\phi(x_t, t)$.

## 2.2 TEXT-TO-3D GENERATION WITH SCORE DISTILLATION

To enable text-to-3D generation using a text-to-image prior, Poole et al. (2022) proposed aligning the distribution of rendered images from an optimizable 3D representation $\theta$ with the text-to-image distribution $p_t(\boldsymbol{x}_t|y)$ generated by a pretrained diffusion model. Let $\Theta$ represent the space of scene parameters, $\mathbb{R}^c$ the space of poses, and $\mathbb{R}^d$ the space of images. Given a pose $c$ and a differentiable renderer $\boldsymbol{g}(\cdot, \cdot) : \Theta \times \mathbb{R}^c \to \mathbb{R}^d$, the distribution of the rendered image is computed through the forward process:

$$q_t^\theta(\boldsymbol{x}_t|c) = \int q_t^\theta(\boldsymbol{x}_0|c)q_t(\boldsymbol{x}_t|\boldsymbol{x}_0)d\boldsymbol{x}_0, \tag{3}$$

where $\boldsymbol{x}_0 = \boldsymbol{g}(\theta, c)$ is the rendered image. The 3D representation $\theta$ is then optimized using a weighted probability density distillation loss:

$$\min_{\theta \in \Theta} \mathcal{L}_{\text{SDS}}(\theta) = \mathbb{E}_{t,c}\left[\left(\frac{\sigma_t}{\alpha_t}\right)\omega(t)D_{\text{KL}}(q_t^\theta(\boldsymbol{x}_t|c) \| p_t(\boldsymbol{x}_t|y^c))\right], \tag{4}$$

where $\boldsymbol{x}_t = \alpha_t \boldsymbol{g}(\theta, c) + \sigma_t \boldsymbol{\epsilon}$, and $y^c$ is a text prompt corresponding to the pose. Since $q_0^\theta(\boldsymbol{x}_0|c)$ is a Dirac distribution $q_0^\theta(\boldsymbol{x}_0|c) = \delta(\boldsymbol{x}_0 - \boldsymbol{g}(\theta, c))$ (Wang et al., 2024a), the gradient of Eq. 4 can be simplified through reparameterization (Ho et al., 2020) as:

$$\nabla_\theta \mathcal{L}_{\text{SDS}}(\theta) = \mathbb{E}_{t,\boldsymbol{\epsilon},c}\left[\omega(t)\left(\epsilon_{\text{pretrain}}(\boldsymbol{x}_t, t, y^c) - \boldsymbol{\epsilon}\right)\frac{\partial \boldsymbol{g}(\theta, c)}{\partial \theta}\right], \tag{5}$$

where $\epsilon \sim \mathcal{N}(0, I)$ and $\epsilon_{\text{pretrain}}$ is the pretrained diffusion denoiser. During optimization, Eq. 5 often results in mode-seeking behavior toward the text-to-image distribution $p_t(\boldsymbol{x}_t|y^c)$, which causes over-smoothness and over-saturation in the generated 3D scene.

To address these issues, Wang et al. (2024b) expanded the point estimate of 3D parameters into a more expressive distribution $\mu(\theta|y)$ by introducing multiple particles $\{\theta^i\}_{i=1}^n$. This extends the simple Gaussian distribution $q_t^\theta(\boldsymbol{x}_t|c)$ in SDS to a more complex distribution:

$$q_t^\mu(\boldsymbol{x}_t|c, y) = \int q_0^\mu(\boldsymbol{x}_0|c, y)p_{t0}(\boldsymbol{x}_t|\boldsymbol{x}_0)d\boldsymbol{x}_0. \tag{6}$$

The distribution $\mu$ is then optimized using a variational score distillation (VSD) objective:

$$\mu^* = \arg\min_\mu \mathbb{E}_{t,c}\left[\left(\frac{\sigma_t}{\alpha_t}\right)\omega(t)D_{\text{KL}}(q_t^\mu(\boldsymbol{x}_t|c, y) \| p_t(\boldsymbol{x}_t|y^c))\right]. \tag{7}$$

To solve this, VSD employs particle-based variational inference (Chen et al., 2018; Liu & Wang, 2016) and fine-tunes an additional U-Net (Ronneberger et al., 2015), $\epsilon_\phi$, using LoRA (Hu et al., 2021). The fine-tuning is formulated as:

$$\min_\phi \sum_{i=1}^n \mathbb{E}_{t\sim\mathcal{U}[0,T],\boldsymbol{\epsilon}\sim\mathcal{N}(\boldsymbol{0},\boldsymbol{I}),c\sim p(c)}\left[\|\epsilon_\phi(\alpha_t \boldsymbol{g}(\theta^{(i)}, c) + \sigma_t\boldsymbol{\epsilon}, t, c, y) - \boldsymbol{\epsilon}\|_2^2\right]. \tag{8}$$

Finally, the gradient for each particle $\theta^i$ is computed as:

$$\nabla_\theta \mathcal{L}_{\text{VSD}}(\theta) = \mathbb{E}_{t,\boldsymbol{\epsilon},c}\left[\omega(t)\left(\epsilon_{\text{pretrain}}(\boldsymbol{x}_t, t, y^c) - \epsilon_\phi(\boldsymbol{x}_t, t, c, y)\right)\frac{\partial \boldsymbol{g}(\theta, c)}{\partial \theta}\right]. \tag{9}$$

## 3 METHOD

The primary goal of our *RecDreamer* is to mitigate the Multi-Face Janus problem through rectification of underlying data distribution in the pre-trained diffusion models. In the following sections, we will first theoretically illustrate the idea of how we rectify the data density via an auxiliary function to ensure a uniform pose distribution (Sec. 3.1). Based on the former theoretical analysis, we introduce a *uniform score distillation* approach for optimizing 3D representations in aligning with the rectified distribution (Sec. 3.2). Furthermore, a series of designed components for implementing the auxiliary function is detailly discussed in Sec. 3.3, including a pose classifier, approximation of the posterior distribution of pose, and estimation of pose-relevant statistics.

### 3.1 RECTIFICATION OF DATA DISTRIBUTION

To directly analyze the relationship between data and pose, we eliminate redundant variables and simplify the text-conditioned probability $p_t(\boldsymbol{x}_t|y)$ to an unconditional density $p(\boldsymbol{x})$, removing the influence of the time step. We denote the data with a general variable $\boldsymbol{x}$. Assuming that $p(\boldsymbol{x}, c)$ represents the joint distribution, the pose distribution can be expressed as $p(c) = \int p(\boldsymbol{x}, c)\mathrm{d}\boldsymbol{x} = \int p(\boldsymbol{x})p(c|\boldsymbol{x})\mathrm{d}\boldsymbol{x}$, which is not a uniform distribution. To mitigate this bias, we frame the simplified problem as follows: given the data distribution $p(\boldsymbol{x})$ and the target attribute distribution $f(c)$, *how can we adjust $p(\boldsymbol{x})$ to a new distribution $\tilde{p}(\boldsymbol{x})$ such that $\tilde{p}(c) = \int \tilde{p}(\boldsymbol{x})p(c|\boldsymbol{x})\mathrm{d}\boldsymbol{x} = f(c)$ holds.*

By introducing a weighting function to the joint probability $p(\boldsymbol{x}, c)$, we establish that the original data density can be adjusted as follows.

**Theorem 1** (Proof in Appendix B.4). *Let $p(\boldsymbol{x})$ denote the data density, $p(c|\boldsymbol{x})$ the conditional distribution of the attribute $c$ given data $\boldsymbol{x}$, and $p(c)$ the marginal distribution of $c$ induced by $p(\boldsymbol{x})$. Given a target distribution $f(c)$ for the attribute $c$, we can construct a new data density $\tilde{p}(\boldsymbol{x})$ such that the marginal distribution of $c$ under $\tilde{p}(\boldsymbol{x})$ matches the target distribution $f(c)$. This new density is given by:*

$$\tilde{p}(\boldsymbol{x}) = p(\boldsymbol{x}) \int \frac{f(c)}{p(c)} p(c|\boldsymbol{x})\, dc. \tag{10}$$

Theorem 1 reveals that the new data density that features a uniformly distributed marginal $f(c)$ can be computed by the original data distribution and an auxiliary function. Furthermore, Theorem 1 can be naturally extended to conditional distributions, as demonstrated in Corollary 2 (see Appendix B.4). So far, we have derived the rectified distribution for clean images, $\tilde{p}(\boldsymbol{x}_0|y)$.

However, since score distillation operates in the noise space, our ultimate goal is to reach the rectified density of the noisy data. Given the transition $p_t(\boldsymbol{x}_t|y) = \int p_0(\boldsymbol{x}_0|y)p_{t0}(\boldsymbol{x}_t|\boldsymbol{x}_0)\mathrm{d}\boldsymbol{x}_0$ where $p_{t0}(\boldsymbol{x}_t|\boldsymbol{x}_0) = \mathcal{N}(\boldsymbol{x}_t|\alpha_t\boldsymbol{x}_0, \sigma_t^2\boldsymbol{I})$, we prove that the rectified distributions for any time step share a unified form, as presented in the following theorem.

**Theorem 2** (Proof in Appendix B.4). *For any $t \sim \mathcal{U}[0, T]$, the rectified density of $\boldsymbol{x}_t$ is given by:*

$$\tilde{p}_t(\boldsymbol{x}_t|y) = p(\boldsymbol{x}_t|y) \int \frac{f(c|y)}{p_t(c|y)} p(c|\boldsymbol{x}_t, y)dc. \tag{11}$$

Theorem 2 reveals that the noisy density of the rectified text-to-image distribution can be expressed as the original noisy density multiplied by an auxiliary function, denoted as $r(\boldsymbol{x}_t|y)$. Specifically, $r(\boldsymbol{x}_t|y) = \int \frac{f(c|y)}{p_t(c|y)} p(c|\boldsymbol{x}_t, y)dc$.

### 3.2 UNIFORM SCORE DISTILLATION

We now return to the original variational distillation problem. First, we define a set of 3D representations $\{\theta^i\}_{i=0}^n$, also named particles in the later gradient flow simulation. Given the distribution $\mu(\theta|y)$ composed of the set $\{\theta^i\}_{i=0}^n$, the camera pose $c$, and the text prompt $y$, the distribution of noisy rendered images is computed as $q_t^\mu(\boldsymbol{x}_t|c, y) = \int q_0^\mu(\boldsymbol{x}_0|c, y)p_{t0}(\boldsymbol{x}_t|\boldsymbol{x}_0)\mathrm{d}\boldsymbol{x}_0$, where $\boldsymbol{x}_0 = \boldsymbol{g}(\theta, c)$. Given the rectified distribution $\tilde{p}_t(\boldsymbol{x}_t|y)$, the objective is as follows:

$$\min_\mu \mathbb{E}_{t,c} \left[ (\sigma_t/\alpha_t)\omega(t)D_{\mathrm{KL}}(q_t^\mu(\boldsymbol{x}_t|c, y) \parallel \tilde{p}_t(\boldsymbol{x}_t|y)) \right]. \tag{12}$$

We refer to this as *uniform score distillation* (USD), as it seeks to approximate the score of the rectified distribution, which is uniformly distributed across the camera poses. To optimize the particles, we derive a corollary based on Theorem 2 from VSD (Wang et al., 2024b):

**Corollary 1** (Corollary to Theorem 2 from VSD). *For Wasserstein gradient flow minimizing Eq. 12, the gradient for the particles is given by:*

$$\nabla_\theta \mathcal{L}_{USD} = \nabla_\theta \mathcal{L}'_{VSD}(\theta) - \mathbb{E}_{t,\boldsymbol{\epsilon},c} \left[ \omega(t)\frac{\sigma_t}{\alpha_t}\nabla_\theta \log r(\boldsymbol{x}_t|y) \right], \tag{13}$$

*where*

$$\nabla_\theta \mathcal{L}'_{VSD} = \mathbb{E}_{t,\boldsymbol{\epsilon},c} \left[ \omega(t)\left(\boldsymbol{\epsilon}_{pretrain}(\boldsymbol{x}_t, t, y) - \boldsymbol{\epsilon}_\phi(\boldsymbol{x}_t, t, c, y)\right) \frac{\partial \boldsymbol{g}(\theta, c)}{\partial \theta} \right], \tag{14}$$

*and $\boldsymbol{x}_t = \alpha_t\boldsymbol{g}(\theta, c) + \sigma_t\boldsymbol{\epsilon}$.*

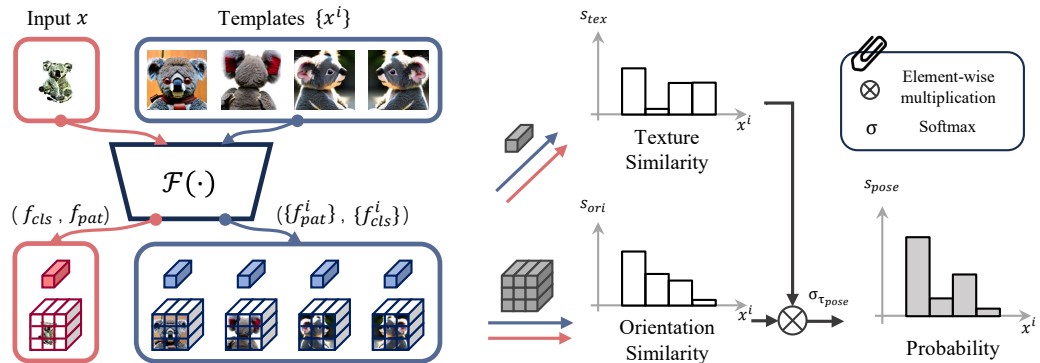

Figure 2: The architecture of our classifier combines orientation and texture similarities in a differential "and-gate" manner. Orientation similarity is evaluated using a patch-matching distance metric, while texture similarity is calculated via cosine similarity of the $[cls]$ token.

Since the rectification algorithm is based on reweighting the sub-distributions, it cannot generate content that was not present in the original distribution. To ensure that $y$ provides a comprehensive distribution including contents of multiple perspectives, we detail the construction techniques in the Appendix B.3.

In the optimization process, we follow VSD by iteratively optimizing the U-Net $\epsilon_\phi$ and the particles $\{\theta^i\}_{i=0}^n$ using Eq. 8 and Eq. 12.

## 3.3 RECDREAMER

The previous sections derive the analytical solution of the rectified distribution and introduce a parameter optimization scheme based on score distillation. To apply this scheme to optimize the 3D scene, we must also compute the rectification function $r(\boldsymbol{x}_t|y)$. We design an effective *classifier to accurately categorize image poses*. Finally, we account for the effects of noisy states and *estimate the posterior distribution of noisy images and its expected value*.

**Discretization of the pose space.** The crux to computing the auxiliary function $r(\boldsymbol{x}_t|y)$ lies in estimating both $p(c|\boldsymbol{x}_t, y)$ and $p_t(c|y)$. Since $p_t(c|y) = \mathbb{E}_{\boldsymbol{x}_t \sim p(\boldsymbol{x}_t|y)} p(c|\boldsymbol{x}_t, y)$ is a term that depends on $p(c|\boldsymbol{x}_t, y)$, we begin by analyzing $p(c|\boldsymbol{x}_t, y)$, which can be interpreted as a pose estimator of noisy images. However, obtaining an estimator for noisy images requires additional data and fine-tuning. To address this, we relate the noisy predictor to the clean predictor by following the DPS (Chung et al., 2022) formulation: $p(c|\boldsymbol{x}_t, y) = \int p(\boldsymbol{x}_0|\boldsymbol{x}_t, y) p(c|\boldsymbol{x}_0, y) d\boldsymbol{x}_0$. Thus, we prioritize the design of a clean estimator $p(c|\boldsymbol{x}_0, y)$ before tackling the noisy case.

Instead of explicitly estimating the camera's extrinsic parameters, we propose modeling a simplified pose by categorizing the images into broad pose categories, such as "front", "back", "left" and "right". In the context of USD, these global categories help maintain a rough balance between different poses and promote 3D consistency. Accordingly, we define the auxiliary function in a discrete form as follows: $r_\xi(\boldsymbol{x}_t|y) = \sum_{\bar{c}} \frac{f(\bar{c}|y)}{p_t(\bar{c}|y)} p_\xi(\bar{c}|\boldsymbol{x}_t, y)$, where $\bar{c}$ represents the discrete pose category, $f(\bar{c}|y) \sim \mathcal{U}\{\bar{c}_i\}_{i=0}^k$, and $p_\xi$ is the parameterized classifier.

**Pose classifier.** Building on this formulation, our goal is to create a lightweight pose classifier without the need for training. To achieve this, we propose a matching-based pose classifier that leverages a pretrained feature extractor and user-provided image templates for each category. Given an input image, the class probabilities are computed by assessing the similarity between the input and the templates. Empirically, the main challenge is distinguishing between 2D orientations (i.e., "left-middle-right") and classifying textures (i.e., "front-back").

To address this, we compute the overall similarity by combining orientation similarity and texture similarity in a differential "and-gate" manner. The pipeline of our classifier is shown in Fig. 2. Drawing inspiration from dense matching techniques (Zhang et al., 2024a;b), we propose using a patch-matching distance metric to evaluate orientation similarity. Texture similarity is determined by

---

**Algorithm 1** Uniform Score Distillation

---

**Require:** A pretrained diffusion model $\epsilon_{pretrain}$, a noise predictor $\epsilon_\phi$ with optimizable parameters $\phi$, a set of particles $\{\theta^i\}_{i=0}^n$, a text prompt $y$, learning rates $\eta_1$ and $\eta_2$, a rectify function $r_\xi$ and a classifier $p_\xi(\bar{c}|\boldsymbol{x}_t, y)$ parameterized by $\xi$, the number of discrete pose categories $n_{\bar{c}}$, the number of time steps $n_{\bar{t}}$, EMA update rate $\alpha_{ema}$.
    Initialize the EMA probabilities $\{\bar{p}_t(\bar{c}|y)\}_{t=0}^{n_t}$, with $\bar{p}_t(\bar{c}|y) = 1/n_{\bar{c}}$.

1: **while** not converged **do**
2:     Randomly sample $\{\theta^i\}_{i=0}^n$ and $c$, render the image $\boldsymbol{x}_0 = \boldsymbol{g}(\theta, c)$.
3:     Apply a forward step $\boldsymbol{x}_t = \mathcal{N}(\boldsymbol{x}_t|\alpha_t\boldsymbol{x}_0, \sigma_t^2\boldsymbol{I})$
4:     $\theta \leftarrow \theta - \eta_1 \mathbb{E}_{t,\boldsymbol{\epsilon},c}\left[\omega(t)\left(\boldsymbol{\epsilon}_{\text{pretrain}}(\boldsymbol{x}_t, t, y) - \boldsymbol{\epsilon}_\phi(\boldsymbol{x}_t, t, c, y)\right)\frac{\partial\boldsymbol{g}(\theta,c)}{\partial\theta}\right]$
          $+\eta_1\mathbb{E}_{t,\boldsymbol{\epsilon},c}\left[\omega(t)\frac{\sigma_t}{\alpha_t}\nabla_\theta\log r_\xi(\boldsymbol{x}_t|y)\right]$
5:     $\bar{p}_t(\bar{c}|y) \leftarrow \alpha_{ema}p_\xi(\bar{c}|\boldsymbol{x}_t, y) + (1 - \alpha_{ema})\bar{p}_t(\bar{c}|y)$
6:     $\phi \leftarrow \phi - \eta_2\nabla_\phi\mathbb{E}_{t,\epsilon}||\boldsymbol{\epsilon}_\phi(\boldsymbol{x}_t, t, c, y) - \boldsymbol{\epsilon}||_2^2$.
7: **end while**
8: **return**

---

calculating the cosine similarity of the $[cls]$ token between the input and template images. Orientation and texture similarities are then multiplied after normalization. Finally, the combined similarity is normalized using a low-temperature softmax function (Goodfellow et al., 2016). For more details on the patch-matching distance and the architecture, please refer to Appendix B.1.

**Estimating** $p(c|\boldsymbol{x}_t, y)$ **and** $p_t(c|y)$**.** By establishing the calculation of $p(c|\boldsymbol{x}_0, y)$ with a plug-and-play pose classifier, we can now introduce the computation of $p(c|\boldsymbol{x}_t, y)$ and $p_t(c|y)$. To compute $p(c|\boldsymbol{x}_t, y) = \int p(\boldsymbol{x}_0|\boldsymbol{x}_t, y)p(c|\boldsymbol{x}_0, y)d\boldsymbol{x}_0$, we follow DPS (Chung et al., 2022) by replacing the calculation of probability with expectation $\mathbb{E}_{\boldsymbol{x}_0 \sim p(\boldsymbol{x}_0|\boldsymbol{x}_t, y)}p(c|\boldsymbol{x}_0, y)$ and further approximating the expectation with Tweedie's formula (Robbins, 1992). Formally, $p(c|\boldsymbol{x}_t, y) \approx p(c|\hat{\boldsymbol{x}}_0, y)$, where $\hat{\boldsymbol{x}}_0 = (\boldsymbol{x}_t - \sigma_t\epsilon_{pretrain}(\boldsymbol{x}_t, t, y))/\alpha_t$. Beyond this approximation, we provide an on-the-fly estimate of the marginal density $p_t(c|y)$, avoiding any form of distribution estimation (Robert, 1999). Concretely, since $p_t(c|y)$ is the expected value of $p(c|\boldsymbol{x}_t, y)$ over $\boldsymbol{x}_t$, we update a distribution $\bar{p}_t(\bar{c}|y)$ using exponential moving average (EMA) of $p(c|\boldsymbol{x}_t, y)$ during optimization, with an update rate $\alpha_{ema}$, to approximate $p(c|\boldsymbol{x}_t, y)$. To enable the in-time estimate of the current pose distribution, we propose a time-interval EMA to capture the distribution. Technical details are left in Appendix B.2.

The proposed scheme allows for the accurate estimation of the auxiliary function $r_\xi$, facilitating the adjustment of the initial distribution so that the sampling results align with the assumption of a uniform pose distribution. The implementation of uniform score distillation is presented in Algorithm 1, and we refer to this systematic approach as *RecDreamer*.

## 4 EXPERIMENTS

### 4.1 EXPERIMENT SETTINGS

To evaluate the performance of USD, we selected 22 prompts describing various objects for comparison experiments. The comparison involves three baseline methods (SDS (Poole et al., 2022), SDS-Bridge (McAllister et al., 2024), and VSD (Wang et al., 2024b)), and three open-source methods designed to address the Multi-Face Janus problem (PerpNeg (Armandpour et al., 2023), Debiased-SDS (Hong et al., 2023), and ESD (Wang et al., 2024a)). We introduce several metrics to assess both the quality of the generated outputs and the severity of the Multi-Face Janus problem. For VSD and USD, we optimize a single particle (*i.e.*, a 3D representation (Mildenhall et al., 2021; Müller et al., 2022)) for score distillation. Additionally, for each prompt, we include auxiliary descriptions to ensure the text-to-image distribution includes the side and back sub-distributions, satisfying the assumption that $p(c) > 0$ in Lemma 1 (see Appendix B.3 for more discussion).

### 4.2 METRICS

We evaluate our approach through three complementary metrics. The Fréchet Inception Distance (Heusel et al., 2017) assesses generation fidelity, while categorical entropy measures quantify

Table 1: Quantitative comparison. The best and second-best results are highlighted in **bold** and underlined, respectively. While these metrics provide valuable insights, they may not fully capture all performance aspects. For a comprehensive evaluation, please refer to the qualitative comparisons and additional experiments in Appendix C.

| Method | FID ($\downarrow$) | uFID ($\downarrow$) | cEnt ($\uparrow$) | pEnt ($\uparrow$) | CLIP ($\downarrow$) |
|---|---|---|---|---|---|
| SDS (Poole et al., 2022) | 204.81 | 205.66 | 1.0235 | 1.1542 | 0.6966 |
| Debiased-SDS (Hong et al., 2023) | 219.46 | 218.83 | 1.0171 | 1.0609 | 0.7251 |
| PerpNeg (Armandpour et al., 2023) | 203.01 | 203.45 | 1.0348 | 1.0390 | 0.7076 |
| ESD (Wang et al., 2024a) | 187.31 | 188.13 | 1.0271 | 1.0928 | 0.6871 |
| SDS-Bridge (McAllister et al., 2024) | 230.87 | 229.41 | 1.0278 | 1.0932 | 0.7250 |
| VSD (Wang et al., 2024b) | 168.19 | 169.66 | 1.0276 | 1.0676 | **0.6807** |
| USD | **165.97** | **165.25** | **1.0375** | **1.2488** | 0.6842 |

distributional bias. Additionally, CLIP (Radford et al., 2021) scores measure the alignment between generated scenes and their corresponding text prompts. Details are left in Appendix C.1.

**Fréchet Inception Distance (FID).** FID evaluates generation quality by comparing two distribution pairs. We compute standard FID against a base diffusion model (60 images per prompt) and unbiased FID (uFID in Table 1) against its pose-balanced version (by annotating and resampling the generated images). For each method, we render 5 images per scene from uniform viewpoints to form the rendered image set for evaluation.

**Categorical Entropy.** We evaluate 3D consistency by quantifying the Multi-Face Janus Problem through classifier predictions. Inconsistent scenes show similar classification probabilities across viewpoints with a bias toward canonical poses, while consistent scenes produce diverse viewpoint-dependent probabilities. We measure this using the entropy of averaged classification probabilities, with higher entropy indicating better consistency. We use two methods: a CLIP-based classifier with directional text descriptions and our proposed pose classifier. The metrics are marked as "cEnt" and "pEnt" in Table 1. For each method, we render 10 images per scene from uniform viewpoints to calculate the entropy.

**CLIP Score.** Following ESD (Wang et al., 2024a), we evaluate text-image alignment by computing CLIP scores between text descriptions and the scenes' corresponding rendered images.

### 4.3 COMPARISONS

**Quantitative Evaluation.** As shown in Table 1, our method outperforms other baselines concerning the measures for generation quality. However, the limited test set size and comparable texture quality between our method and VSD make it challenging to fully quantify the impact of geometric consistency through these metrics alone. We provide more details in qualitative comparison and Appendix C. In terms of diversity measures, our method achieves higher entropy scores in both CLIP-based categorization (cEnt) and pose classification (pEnt), indicating that USD effectively incorporates multi-view information into the 3D representations and mitigates the issue of repetitive patterns across different viewpoints. Regarding text-image alignment, USD shows lower CLIP scores than VSD, as our multi-view approach incorporates back and side views that may not align with prompts describing predominantly frontal features (*e.g.*, back views of a dog versus front-oriented descriptions).

**Qualitative Evaluation.** As shown in Fig. 3, the texture quality achieved by our method is comparable to that of VSD. In terms of geometry, USD demonstrates a reasonable structure, capturing the shapes of different poses and successfully simulating some finer details like bumps. Although some artifacts remain (not as smooth as SDS and its variants), our method maintains a relatively accurate geometry compared to VSD. More results are available in `https://chansey0529.github.io/RecDreamer_proj/`.

### 4.4 ABLATIONS

We provide ablation results in Fig. 4. Since the first stage of training establishes the overall geometry, all subsequent experiments are conducted using only the first stage for comparison.

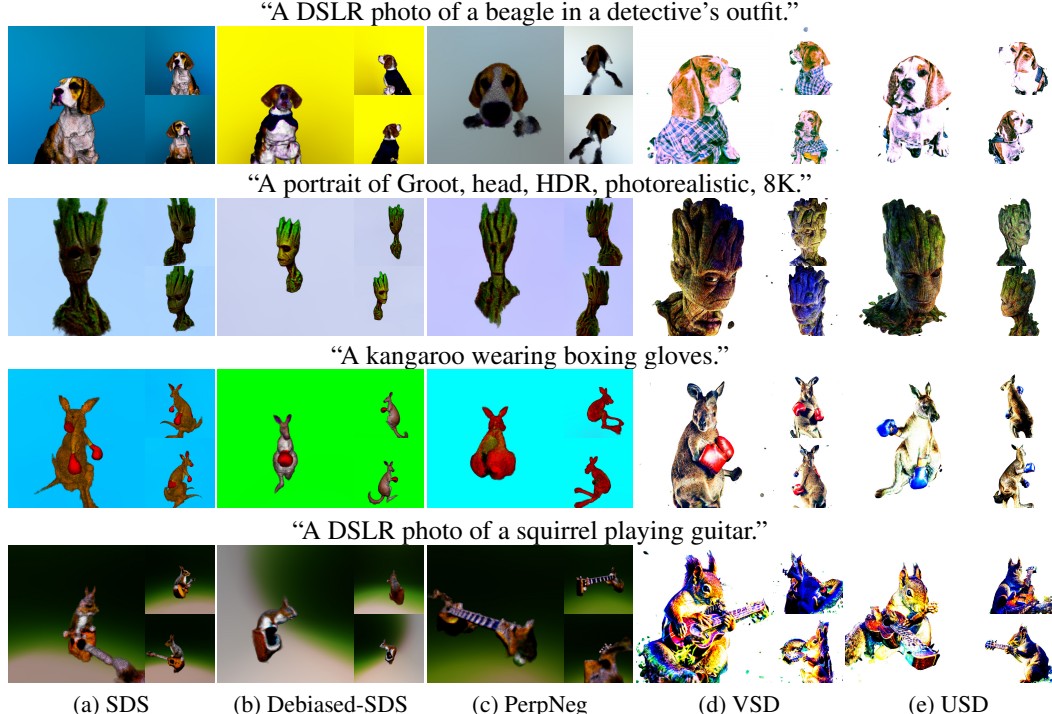

"A DSLR photo of a beagle in a detective's outfit."

"A portrait of Groot, head, HDR, photorealistic, 8K."

"A kangaroo wearing boxing gloves."

"A DSLR photo of a squirrel playing guitar."

(a) SDS     (b) Debiased-SDS     (c) PerpNeg     (d) VSD     (e) USD

Figure 3: Qualitative comparison. The text-to-3D generation results are visualized from three perspectives (front, left, and right side views), illustrating how the rectified distribution in our USD framework effectively mitigates the Multi-Face Janus phenomenon.

**Ablation for** $p(c|\boldsymbol{x}_t, y)$**.** To verify the approximation of $p(c|\boldsymbol{x}_t, y)$, we design a variant that directly estimates the distribution of the noisy images $\boldsymbol{x}$ using a pose classifier. As shown in Fig. 4, the rectification term in this implementation is almost ineffective because the classifier struggles to determine the class of noisy images. This causes the rectification to work only in relatively small time steps for clean images, limiting its impact on the global optimization. This result highlights the necessity of approximating $p(c|\boldsymbol{x}_t, y)$.

**Ablation for** $p(c|y)$**.** Furthermore, to validate the effectiveness of $p(c|y)$, we devise a sampling variant. Instead of predicting the current distribution in real-time using EMA, we sample a batch of images before training and predict the distribution for each interval. Score distillation is then performed based on this fixed pose distribution. However, the results are quite random (the "bear" case is over-rectified and the "zombie" case is under-rectified). This is because there may be a gap between the distribution of score distillation and the sampled distribution, leading to incorrect guidance to generate another bias. Additionally, a fixed rectified distribution is unable to adaptively balance the gradient of the noise predictor and the classifier, therefore may lead to over-adjustment.

In conclusion, we adopt the approximation $p(c|\boldsymbol{x}_t, y) \approx p(c|\hat{\boldsymbol{x}}_0, y)$ and employ dynamic distribution updates via EMA. This ensures an effective simulation of the rectification function values.

### 4.5 COMPONENT ANALYSIS

**Hyperparameter Evaluation.** We analyze the impact of key hyperparameters, such as the update rate of EMA $\alpha_{ema}$ and the number of particles $n_t$. The results indicate that a larger $\alpha_{ema}$ facilitates more responsive updates, allowing for real-time tracking and adjustment of the pose distribution. Additionally, our findings suggest that the back-and-forth time scheduling, as detailed in Appendix B.3, enhances multi-particle optimization. Further specifics can be found in Appendix C.2.

**Additional Experiments.** In addition to the hyperparameter evaluation, we conduct further investigations, detailed in Appendices C, D, and E. Using the annotated pose data, we quantitatively validate the effectiveness of the pose classifier, with ablation studies on texture and orientation scores confirming the robustness of the classifier architecture. Validation experiments on 2D particles pro-

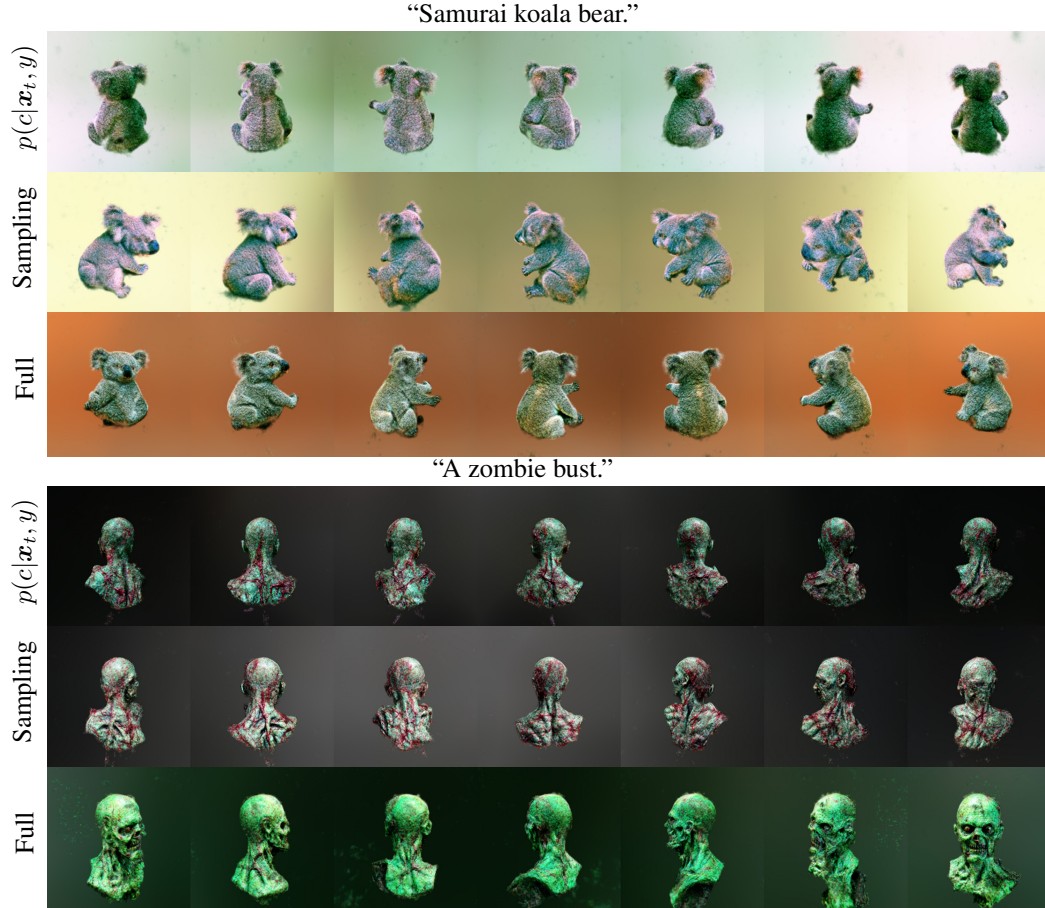

Figure 4: Ablation studies. We construct two variants, $p(c|\boldsymbol{x}_t, y)$ and $p_t(c|y)$ for comparison. The variant $p(c|\boldsymbol{x}_t, y)$ directly predicts the category of noisy images $\boldsymbol{x}_t$, while the sampling-based method, $p_t(c|y)$, estimates the pose distribution by generating multiple samples and predicting the respective categories.

vide an intuitive demonstration of USD's performance. Furthermore, by utilizing RecDreamer, we extend the conditional image generation (Graikos et al., 2022) from one single particle into a multi-particle optimization scheme, enabling more effective control with promising practical applications.

## 5 CONCLUSION

In this paper, we presented RecDreamer, a novel approach to mitigating the Multi-Face Janus problem in text-to-3D generation. Our solution introduces a rectification function to modify the prior distribution, ensuring that the resulting joint distribution achieves uniformity across poses. By expressing the modified data distribution as the product of the original density and the rectification function, we seamlessly integrate this adjustment into the score distillation algorithm. This allows us to derive a particle optimization framework for uniform score distillation. Additionally, we developed a pose classifier and implemented reliable approximations and simulations to enhance the particle optimization process. Extensive experiments on both 2D and 3D synthesis tasks demonstrate the effectiveness of our approach in addressing the Multi-Face Janus problem, resulting in more consistent geometries and textures across different views.

**Limitations.** While our method significantly reduces bias in prior distributions, further exploration of 3D modeling with multi-view priors could improve geometric and texture consistency. Extending our approach through deeper research into conditional control presents another promising avenue for addressing these challenges in future work.

ACKNOWLEDGMENTS

The work is supported by Guangdong Provincial Natural Science Foundation for Outstanding Youth Team Project (No. 2024B1515040010), NSFC Key Project (No. U23A20391), China National Key R&D Program (Grant No. 2023YFE0202700, 2024YFB4709200), Key-Area Research and Development Program of Guangzhou City (No. 202206030007, 2023B01J0022), the Guangdong Natural Science Funds for Distinguished Young Scholars (No. 2023B1515020097), the AI Singapore Programme under the National Research Foundation Singapore (No. AISG3-GV-2023-011), and the Lee Kong Chian Fellowships.

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

# A RELATED WORKS

## A.1 DIFFUSION MODELS FOR TEXT-TO-IMAGE GENERATION

A major challenge in text-to-image generation using diffusion models (Yu et al., 2024; Liu et al., 2024; Xu et al., 2024) is guiding the generative process to reflect the input text accurately. A widely adopted solution is classifier-free guidance (CFG, Ho & Salimans (2022)), which eliminates the need for external classifiers by training a unified model for both unconditional and conditional image generation. During inference, CFG achieves conditional generation by interpolating between the conditional and unconditional scores, effectively guiding the model to match the input text. This method has shown significant success in various text-to-image tasks (Balaji et al., 2022; Nichol et al., 2021; Ramesh et al., 2022). Models like DALL·E 2 and Stable Diffusion (Rombach et al., 2022) have demonstrated exceptional capabilities in producing diverse and complex images, with promising extensions into text-to-3D generation.

## A.2 TEXT-TO-3D GENERATION

Recent advances in text-to-3D generation can be broadly divided into two main approaches. The first approach focuses on directly learning 3D asset distributions from large-scale datasets such as Objaverse (Deitke et al., 2023). Notable models within this category include GET3D (Gao et al., 2022), Point-E (Nichol et al., 2022), Shap-E (Jun & Nichol, 2023), CLAY (Zhang et al., 2024c), and MeshGPT (Siddiqui et al., 2024), all of which leverage extensive 3D data to generate accurate 3D models.

The second approach relies on 2D priors (Jiang et al., 2023) for generating 3D models. Techniques like score distillation are foundational here, as exemplified by DreamFusion/SJC (Poole et al., 2022; Wang et al., 2023a) and ProlificDreamer (Wang et al., 2024b).

Building on these baselines, researchers continue to improve visual quality. Classifier Score Distillation (Yu et al., 2023) reframes the fundamental approach by treating classifier-free guidance as the central mechanism rather than a supplementary component, enabling more realistic synthesis. Noise-Free Score Distillation (Katzir et al., 2023) addresses the over-smoothing problem by eliminating unnecessary noise terms, allowing for effective generation at standard guidance scales. SteinDreamer (Wang et al., 2023b) introduces Stein's identity to reduce gradient variance in score distillation for faster and higher-quality generation. LucidDreamer (Liang et al., 2024) tackles the over-smoothing challenge differently by combining interval score matching with 3D Gaussian Splatting (Kerbl et al., 2023) and deterministic diffusion paths. Most recently, SDS-Bridge (McAllister et al., 2024) enhances the entire pipeline by introducing calibrated sampling based on optimal transport theory and improved distribution estimates.

A separate line of research focuses on improving geometric quality. Magic3D (Lin et al., 2023) enhances output resolution by first generating a coarse 3D hashgrid and subsequently refining it into a mesh. Fantasia3D (Chen et al., 2023) introduces hybrid scene representations and spatially varying BRDF (of Standards & Nicodemus, 1977) for realistic modeling. Other models, such as Dreamtime (Huang et al., 2023) and HiFA (Zhu et al., 2023), concentrate on optimizing time-step sampling to improve texture stability and geometric consistency.

## A.3 ALLEVIATING THE MULTI-FACE JANUS PROBLEM

One of the key challenges when extending text-to-image priors to 3D is the Multi-Face Janus problem, where inconsistencies arise in 3D geometries, especially for objects with multiple faces. To address this, Yi et al. (2023) and Ma et al. (2023) introduce pretrained shape generators that provide geometric priors. DreamCraft3D (Sun et al., 2023) tackles the challenge through a hierarchical framework, combining view-dependent diffusion with Bootstrapped Score Distillation to separate geometry and texture optimization. MVDream (Shi et al., 2023) introduces a dedicated multi-view diffusion model that bridges 2D and 3D domains, enabling few-shot learning from 2D examples. JointDreamer (Jiang et al., 2025) presents Joint Score Distillation with view-aware models as energy functions to ensure coherence by explicitly modeling cross-view relationships. While these methods effectively handle diverse scenarios, certain complex text descriptions can still pose challenges due to inherent limitations in pretrained generators and multi-view generative models.

Beyond introducing basic geometric priors, several methods have aimed to improve control over pose prompts. Debiased-SDS (Hong et al., 2023) tackles text bias by removing words that conflict with pose descriptions, while Perp-Neg (Armandpour et al., 2023) proposes a perpendicular gradient sampling technique to remove undesired attributes from negative prompts. Other works have sought to address pose bias in pretrained models by altering the approximation distribution through pose sampling or entropy constraints.

DreamControl (Huang et al., 2024) approaches the pose bias issue by employing adaptive viewpoint sampling, which adjusts the rendering pose distribution to better mimic the inherent biases of the model. Additionally, ESD (Wang et al., 2024a) demonstrates that the score distillation process degenerates into maximum-likelihood seeking, and proposes an entropic term to introduce diversity across different views, helping to prevent repetitive patterns in 3D generation.

## B   METHODOLOGICAL DETAILS

In this part, we further complement the introduction of the RecDreamer. Appendix B.1 presents the detailed architecture of the pose classifier. B.2 provides a more detailed derivation of the approximation discussed in Sec. 3.3, and an explanation of the EMA update process. In B.3, we present the precise implementation details of RecDreamer. Lastly, the proof of the main theorems is given in B.4.

### B.1   POSE CLASSIFIER

#### B.1.1   OVERVIEW

The classification of pose typically involves two key points: when shapes of input and template images are similar (front and back), texture information is usually used for differentiation. Conversely, when shapes are different (left, center, and right), attention must be given to the features' positions.

As introduced in Sec. 3.3, the classifier mimics an "AND gate" structure, combining texture similarity and orientational similarity between the input and template for computation. Here, texture similarity mainly distinguishes the front and back of objects, while orientational similarity differentiates the three 2D orientations (left, center, and right).

Texture similarity is obtained by calculating the cosine similarity between global features (*i.e.*, $[cls]$ token) derived from the input and template images. However, orientational similarity cannot be easily derived from pretrained feature extractors, as they often use image augmentation techniques (like flipping) during training, leading to similar global features for left and right orientations. To address this issue, we propose the most matching patch distance to measure orientational similarity. The overall process is illustrated in Fig. 2 and formulation is detailed in Appendix B.1.2.

#### B.1.2   ARCHITECTURE

Assume that we have a feature extractor $\mathcal{F}(\cdot)$ that maps an image to the feature space. Given the height $h$ and width $w$ of the features, we denote the global feature and patch features of an input image $\boldsymbol{x} \in \mathbb{R}^d$ as $\boldsymbol{f}_{cls} \in \mathbb{R}^f$ and $\boldsymbol{f}_{pat} \in \mathbb{R}^{h \times w \times f}$, *i.e.*, $\boldsymbol{f}_{cls}, \boldsymbol{f}_{pat} = \mathcal{F}(\boldsymbol{x})$. The features of template images $\{\boldsymbol{x}^i | 0 \leq i < n_p\}$ are given by $\{\boldsymbol{f}_{cls}^i | 0 \leq i < n_p\}$, $\{\boldsymbol{f}_{pat}^i | 0 \leq i < n_p\}$, where $n_p$ is the number of the pose categories. To calculate the texture similarity, we directly calculate the cosine similarity $cos(\cdot, \cdot)$ between the global input features and the template features as follows:

$$\boldsymbol{s}_{tex} = \{s_{tex}^i | 0 \leq i < n_p\},$$
$$s_{tex}^i = cos(\boldsymbol{f}_{cls}, \boldsymbol{f}_{cls}^i). \tag{15}$$

In Eq. 15, we consolidate the per-class texture similarity scores (scalars) $s_{tex}^i$ into a vector representation $\boldsymbol{s}_{tex}$ to streamline notation.

To calculate the orientation similarity, we propose the matching patch distance to evaluate the orientation discrepancy between the input and output images. To concentrate on the main subject, we introduce the binary mask (the calculation of binary mask is introduced in Appendix B.1.3) of both input and template images, denoted as $\boldsymbol{b}$ and $\{\boldsymbol{b}^i | 0 \leq i < n_p\}$, where $\boldsymbol{b} \in \mathbb{R}^{h \times w \times 1}$. We also

distribute a coordinate map for the input and template images based on the binary masks to mark the relevant coordinate, denoted as $m$ and $m^i$. To be specified, the leftmost pixel of the subject is assigned with a value $-0.5$ and the rightmost as $0.5$, where the value of intermediate pixels between the leftmost and rightmost are interpolated from $-0.5$ to $0.5$.

We denote the patch features by $\boldsymbol{f}_{pat}$ and $\boldsymbol{f}^i_{pat}$, the binary masks by $\boldsymbol{b}$ and $\boldsymbol{b}^i_{pat}$, and the coordinate maps by $\boldsymbol{m}$ and $\boldsymbol{m}^i_{pat}$. At a specific patch coordinate $u = (x, y)$, these quantities are written as $(\boldsymbol{f}_u, \boldsymbol{f}^i_u)$, $(b_u, b^i_u)$, and $(m_u, m^i_u)$, respectively. For subject patches (that is, coordinates where the binary mask is 1), we collect the corresponding features and coordinates into the sets $\hat{\boldsymbol{f}}_{pat} = \{\boldsymbol{f}_u \mid b_u = 1\}$ and $\hat{\boldsymbol{m}} = \{m_u \mid b_u = 1\}$. Similarly, for template images, we denote these sets by $\hat{\boldsymbol{f}}^i_{pat}$ and $\hat{\boldsymbol{m}}^i$.

For the input image, the similarities between the foreground patch $\hat{\boldsymbol{f}}_{pat,u}$ and the template patch $\hat{\boldsymbol{f}}^i_{pat,u'}$ is accessed by calculating the L2-norm, i.e., $s_{u,u'} = 1 - \sigma_{\tau_{pat}}(\|\hat{\boldsymbol{f}}_{pat,u} - \hat{\boldsymbol{f}}^i_{pat,u'}\|_2)$, where $\sigma_{\tau_{pat}}(\cdot)$ is a softmax function with temperature. Then, the mean distance between the input and the template is computed by traversing all the patches of the input image and the template image, and we use the negative of the distance to indicate the similarity of orientation:

$$
\boldsymbol{s}_{ori} = \{s^i_{ori} | 0 \le i < n_p\},
$$
$$
s^i_{ori} = 1 - \frac{1}{\|u\|\|u'\|} \sum_u \sum_{u'} s_{u,u'} \|\hat{m}_u - \hat{m}^i_{u'}\|. \tag{16}
$$

Finally, we compute the final probability $\boldsymbol{s}_{pose}$ by:

$$
\boldsymbol{s}_{pose} = \sigma_{\tau_{pose}}(\boldsymbol{s}_{tex} \otimes \boldsymbol{s}_{ori}). \tag{17}
$$

Here, $\sigma_{\tau_{pose}}(\cdot)$ is a softmax function and $\otimes$ signifies element-wise multiplication. In Fig. 5, we demonstrate the distance calculation between a single pixel $u$ from the input image and the pixels $\{u'\}$ in the templates.

### B.1.3 FOREGROUND MASK SEGMENTATION

To segment the foreground subject, we follow DINOv2 (Oquab et al., 2023; Darcet et al., 2023) and apply a Principle Component Analysis (PCA) (Abdi & Williams, 2010) to the patch features $\{\boldsymbol{f}^i_{pat} | 0 \le i < n_p\}$. The PCA algorithm reduces the dimensions of the patch from $\mathbb{R}^{h \times w \times f}$ to $\mathbb{R}^{h \times w \times f'}$, where $f'$ is a small number. We use the first component along the feature channel dimension, as the basis for foreground segmentation. However, since it's unclear whether $> 0$ corresponds to the subject or $< 0$ does after PCA, we introduce the cross-attention feature map from Stable Diffusion (Rombach et al., 2022) for guidance. Concretely, we encode the template image into latent space and use the cross-attention features of the image and prompt at different time steps to obtain the subject area. Subsequently, based on the cross-attention feature map, we can infer whether the positive or negative value of PCA's first component stands for foreground. Note that we do not use cross-attention directly as a segmentation map because this method is more complex and not suitable for efficient segmentation during training. Instead, using PCA for segmentation is a lightweight approach that does not affect training efficiency.

### B.1.4 IMPLEMENTATION OF POSE CLASSIFIER

To minimize the impact on efficiency, we employ the feature extractor "dinov2_vits14". We resize the classifier features to a size $h = w = 16$. The temperature for patch-wise similarity $\tau_{pat}$ is set to $0.01$, while the temperature for pose in Eq. 16 is set to $0.05$ to enhance the distinction between categories. When implementing the classifier (calculating PCA parameters), we apply data augmentations (random noise, random affine, random grayscale, and color jitter) to the template images to achieve robust foreground-background segmentation and classification.

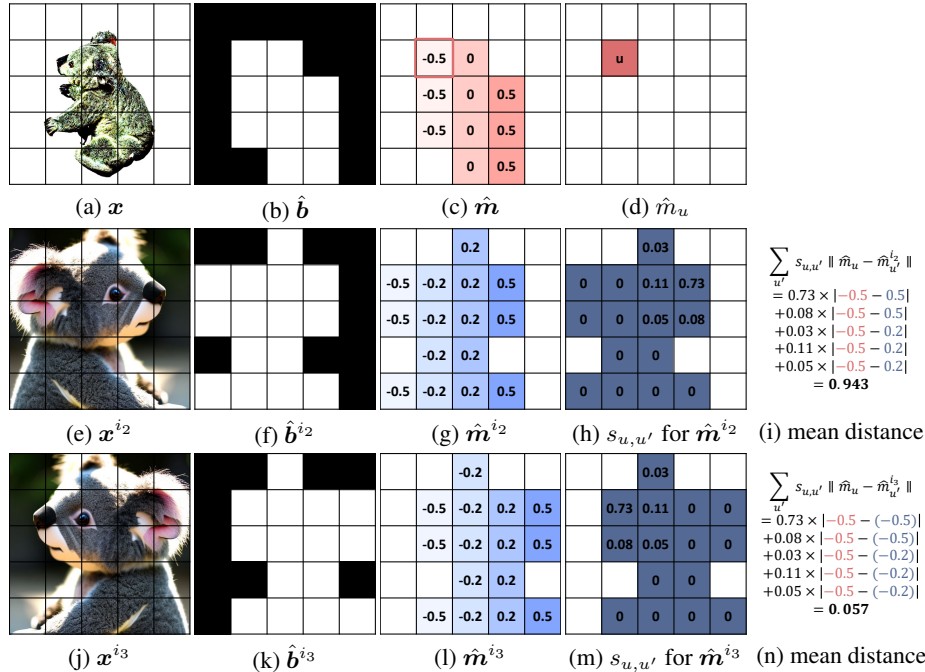

Figure 5: Calculation of orientation distance. (a) is the input image for testing. (e) and (j) are templates showing "facing right" and "facing left" orientations. Each row demonstrates the orientation distance between the specific image patch (d) and the templates. For opposite orientations, the mean distance is larger compared to the similar orientations ((i) v.s. (n)). This feature is important for the classification of pose.

## B.2 CALCULATION OF THE RECTIFICATION FUNCTION

**Approximation of** $p(c|\boldsymbol{x}_t, y)$. In Sec. 3.3, we have demonstrated the approximation of $p(c|\boldsymbol{x}_t, y)$. By expanding the derivation of Eq. 7 of DPS (Chung et al., 2022) to a conditional form, we have:

$$
\begin{aligned}
p(c|\boldsymbol{x}_t, y) &= \int p(\boldsymbol{x}_0|\boldsymbol{x}_t, y)p(c|\boldsymbol{x}_0, \boldsymbol{x}_t, y)d\boldsymbol{x}_0 \\
&= \int p(\boldsymbol{x}_0|\boldsymbol{x}_t, y)p(c|\boldsymbol{x}_0, y)d\boldsymbol{x}_0 \\
&= \mathbb{E}_{\boldsymbol{x}_0 \sim p(\boldsymbol{x}_0|\boldsymbol{x}_t, y)}p(c|\boldsymbol{x}_0, y).
\end{aligned}
\tag{18}
$$

Note that we follow DPS's assumption that pose $c$ is independent to $\boldsymbol{x}_t$. Eq. 18 is approximated by exchanging the calculation of expectation and $p(c|\boldsymbol{x}_0, y)$, *i.e.*, $p(c|\boldsymbol{x}_t, y) \approx p(c|\hat{\boldsymbol{x}}_0, y)$, where $\hat{\boldsymbol{x}}_0 = \mathbb{E}_{\boldsymbol{x}_0 \sim p(\boldsymbol{x}_0|\boldsymbol{x}_t, y)}[\boldsymbol{x}_0]$. By Tweedie's formula, $\hat{\boldsymbol{x}}_0 \approx (\boldsymbol{x}_t - \sigma_t \epsilon_{pretrain}(\boldsymbol{x}_t, t, y))/\alpha_t$.

**Estimate of** $p_t(c|y)$. Another term, $p_t(c|y)$, represents the expectation of $p(c|\boldsymbol{x}_t, y)$ over $\boldsymbol{x}_t$. To estimate $p_t(c|y)$, we employ an exponential moving average (EMA) to iteratively update this term with the values of $p(c|\boldsymbol{x}_t, y)$ during training. Since $p_t(c|y)$ is time-dependent, each time step in the diffusion model requires a corresponding EMA value. However, updating the EMA at each iteration only occurs with a probability of 1/1000, which does not accurately track the current distribution. Fortunately, we observe that the empirical discrete pose probability of $p_t(\bar{c}|y)$ for adjacent time steps are nearly identical (e.g., $p_1(\bar{c}|y)$ and $p_2(\bar{c}|y)$ are almost the same). As a result, the EMA values across multiple time steps can be unified within intervals to enhance efficiency. We denote the number of intervals as $n_t$, and the number of steps within each interval, $n_s$, is calculated as $n_s = T/n_t$, where $T$ is the training step of DDPM. We maintain a list of EMA values for different intervals, $\{v_{ema}^i\}_{i=0}^{n_t}$. The EMA version of the pose probability, $\bar{p}_t(\bar{c}|y)$, is then given by $\bar{p}_t(\bar{c}|y) = v_{ema}^{i=\lfloor t/n_s \rfloor}$. The update of the EMA follows the update rate $\alpha_{ema}$, as described in the following:

$$\bar{p}_t(\bar{c}|y) \leftarrow \alpha_{ema} p_\xi(\bar{c}|\boldsymbol{x}_t, y) + (1 - \alpha_{ema}) \bar{p}_t(\bar{c}|y) \tag{19}$$

Empirically, given $T = 1000$ is the total steps, we set $n_t = 10$ and $n_s = 100$. $\alpha_{ema}$ is set to satisfy that the previous $n_{ema}$ samples has total EMA weights greater than $0.9$.

### B.3 OTHER IMPLEMENTATIONS

Additionally, we implement other tricks to ensure the effectiveness of gradient updates at different time steps.

**Gradient norm.** The optimization dynamics of uniform score distillation present a fundamental challenge: the framework simultaneously processes two conflicting gradient signals (classifier gradients and denoiser gradients) with inherent magnitude disparity. Through empirical measurement, we observe that classifier gradients for rendered images $\boldsymbol{x}_0$ exhibit substantially higher magnitudes compared to denoiser gradients. To resolve this optimization instability, we implement gradient norm alignment - specifically constraining the L2 norm of the classifier gradient to match the denoiser gradient norm. This normalization strategy ensures balanced parameter updates while preserving relative gradient directions.

**Time scheduler.** Furthermore, we utilized a back-and-forth (BNF) time scheduler. The time scheduler constrains the sampling interval for each iteration. Assuming the number of intervals for BNF is denoted as $n_i$, we divide the total number of iterations into $2n_i$ intervals. For the first $n_i$ intervals, the sampled time steps are expanded from $[T * 0.98, T - (T/n_i)]$ to $[T * 0.98, T * 0.02]$. For the last $n_i$ intervals, the sampled time steps are reduced from $[T * 0.98, T * 0.02]$ to $[(T/n_i), T * 0.02]$. Typically, we set $n_i = 2$ for one particle optimization.

**Three-stage optimization.** Similar to VSD, we use a three-stage optimization paradigm. For the first stage, we train the Instant-NGP (Müller et al., 2022) using USD for $15k$ iters. In the second stage, we use SDS for geometric refinement for $15k$ iters. In the third stage, we optimize the texture with USD for $15k$ iters.

**Auxiliary prompts.** Our method is essentially reweighting the subdistribution of a prior distribution, so in the proof of Lemma 1 we assume that $p(c) > 0$. To ensure this condition holds, we augment the original prompt with phrases like "from side view, from back view." to introduce additional pose information. While these auxiliary prompts may not create a perfectly balanced distribution, they guarantee that $p(c) > 0$ by incorporating multiple viewpoints. Our algorithm then adjusts this distribution to achieve uniformity. It is important to note that our use of directional text differs fundamentally from prior work. Previous research has primarily employed directional text for conditional generation, aiming to constrain the model to produce content that strictly aligns with the provided text. In contrast, our approach leverages directional prompts to expand the model's distribution, allowing it to capture a more diverse range of information.

### B.4 PROOF OF MAIN THEOREM

#### B.4.1 PROOF OF THEOREM 1

Since two marginal distributions $p(\boldsymbol{x})$ and $p(c)$ are involved, we first study their joint distribution $p(\boldsymbol{x}, c)$. We introduce the weighting function $w(c)$ to correct $p(\boldsymbol{x}, c)$ so that the rectified marginal distribution obeys the target distribution $f(c)$. The rectified joint distribution is given by the following lemma.

**Lemma 1.** *Given the original joint distribution $p(\boldsymbol{x}, c)$, where $p(c)$ is the marginal distribution of $c$, and $p(c) \neq 0$, and a target marginal distribution $f(c)$, we can rectify $p(c)$ to $f(c)$ by introducing a weighting function $w(c) = \frac{f(c)}{p(c)}$. The corrected joint distribution $\tilde{p}(\boldsymbol{x}, c)$ is then given by:*

$$\tilde{p}(\boldsymbol{x}, c) = w(c)p(\boldsymbol{x}, c) = \frac{f(c)}{p(c)}p(\boldsymbol{x}, c). \tag{20}$$

*Proof of Lemma 1.* To adjust the marginal distribution $p(c)$ to the target distribution $f(c)$, we apply a weighting function $w(c)$ to the original joint distribution following importance sampling. The new

joint density is given by:

$$\tilde{p}(\boldsymbol{x}, c) = w(c)p(\boldsymbol{x}, c). \tag{21}$$

The marginal distribution of $c$ under $\tilde{p}(\boldsymbol{x}, c)$ is:

$$\tilde{p}(c) = \int \tilde{p}(\boldsymbol{x}, c)d\boldsymbol{x} = \int w(c)p(\boldsymbol{x}, c)d\boldsymbol{x} = w(c)p(c). \tag{22}$$

To satisfy $\tilde{p}(c) = f(c)$, we set $w(c) = \frac{f(c)}{p(c)}$. Substituting this into the expression for $\tilde{p}(\boldsymbol{x}, c)$ gives Eq. 20. Since $f(c)$, $p(c)$, and $p(\boldsymbol{x}, c)$ are non-negative, $\tilde{p}(\boldsymbol{x}, c) \geq 0$. To validate normalization, we compute:

$$\int \int \tilde{p}(\boldsymbol{x}, c)d\boldsymbol{x}dc = \int \frac{f(c)}{p(c)} \left( \int p(\boldsymbol{x})p(c|\boldsymbol{x})d\boldsymbol{x} \right) dc = 1. \tag{23}$$

This confirms that $\tilde{p}(\boldsymbol{x}, c)$ is a valid probability distribution, which completes the proof. $\square$

Below we provide proof of Theorem 1.

*Proof of Theorem 1.* According to Lemma 1, the rectified joint distribution $\tilde{p}(\boldsymbol{x}, c)$ satisfies that the marginal distribution $\tilde{p}(c) = f(c)$. The rectified data density $\tilde{p}(\boldsymbol{x})$ is obtained by marginalizing $\tilde{p}(\boldsymbol{x}, c)$ over $c$ as follow:

$$\tilde{p}(\boldsymbol{x}) = \int \tilde{p}(\boldsymbol{x}, c)dc = \int \frac{f(c)}{p(c)}p(\boldsymbol{x})p(c|\boldsymbol{x})dc = p(\boldsymbol{x}) \int \frac{f(c)}{p(c)}p(c|\boldsymbol{x})dc \tag{24}$$

This completes the derivation of $\tilde{p}(\boldsymbol{x})$. $\square$

### B.4.2 COROLLARY OF THEOREM 1

We generalize the conclusions of Theorem 1 to conditional distributions as follows.

**Corollary 2.** *For the conditional case, we can extend the result to $\tilde{p}(\boldsymbol{x}|y)$ as:*

$$\tilde{p}(\boldsymbol{x}|y) = p(\boldsymbol{x}|y) \int \frac{f(c|y)}{p(c|y)}p(c|\boldsymbol{x}, y)dc. \tag{25}$$

*This follows directly from the general form by conditioning on $y$.*

*Proof of Corollary 2.* Analogous to Lemma 1, we aim to rectify the conditional marginal distribution $p(c|y)$ to the target distribution $f(c|y)$. By $\tilde{p}(c|y) = \int p(c, \boldsymbol{x}|y)dx = w(c|y)p(c|y) = f(c|y)$, we derive $w(c|y) = \frac{f(c|y)}{p(c|y)}$. The rectified distribution $\tilde{p}(\boldsymbol{x}|y)$ is then expressed by integrating over $c$ as follows:

$$\tilde{p}(\boldsymbol{x}|y) = \int \tilde{p}(c, \boldsymbol{x}|y)dc = \int \frac{f(c|y)}{p(c|y)}p(c, \boldsymbol{x}|y)dc = p(\boldsymbol{x}|y) \int \frac{f(c|y)}{p(c|y)}p(c|\boldsymbol{x}, y)dc. \tag{26}$$

This completes the derivation of $\tilde{p}(\boldsymbol{x}|y)$. $\square$

### B.4.3 PROOF OF THEOREM 2

We provide the proof for deriving the rectified density for different time steps.

**Remark 1.** *Our primary objective is to obtain the rectified density at $t = 0$, i.e., $\tilde{p}_0(\boldsymbol{x}_0 \mid y)$, without imposing a uniform distribution across all other noise states. Consequently, for any $t > 0$, the density $\tilde{p}_t(\boldsymbol{x}_t|y)$ should be derived from a transition from $\tilde{p}_t(\boldsymbol{x}_0|y)$, as detailed in the following proof.*

*Proof of Theorem 2.* First, we consider the $t = 0$. According to Corollary 2, we have:

$$\tilde{p}_0(\boldsymbol{x}_0|y) = p_0(\boldsymbol{x}_0|y) \int \frac{f(c|y)}{p_0(c|y)}p(c|\boldsymbol{x}_0, y)dc. \tag{27}$$

For any $t \in [1, T]$, the probability of noisy images is given by:

$$
\begin{aligned}
\tilde{p}_t(\boldsymbol{x}_t|y) &= \int \tilde{p}_0(\boldsymbol{x}_0|y) p_{t0}(\boldsymbol{x}_t|\boldsymbol{x}_0) d\boldsymbol{x}_0 \\
&= \int \left[ \int f(c|y) p(\boldsymbol{x}_0|c, y) dc \right] p_{t0}(\boldsymbol{x}_t|\boldsymbol{x}_0) d\boldsymbol{x}_0 \\
&\stackrel{(a)}{=} \int f(c|y) \left[ \int p(\boldsymbol{x}_0|c, y) p_{t0}(\boldsymbol{x}_t|\boldsymbol{x}_0) d\boldsymbol{x}_0 \right] dc \\
&\stackrel{(b)}{=} \int f(c|y) \left[ \int p(\boldsymbol{x}_0|c, y) p_{t0}(\boldsymbol{x}_t|\boldsymbol{x}_0, c, y) d\boldsymbol{x}_0 \right] dc \\
&= \int f(c|y) \left[ \int p(\boldsymbol{x}_t, \boldsymbol{x}_0|c, y) d\boldsymbol{x}_0 \right] dc \\
&= \int f(c|y) p(\boldsymbol{x}_t|c, y) dc \\
&= p_t(\boldsymbol{x}_t|y) \int \frac{f(c|y)}{p_t(c|y)} p(c|\boldsymbol{x}_t, y) dc,
\end{aligned}
\tag{28}
$$

where (a) is according to Fubini's theorem, (b) is based on the that the forward process proceeds according to the original scheme, unaffected by text or pose conditions. By combining Eq. 27 and Eq. 28, we conclude that for any $t \in [0, T]$, Eq. 11 holds, completing the proof. $\square$

### B.4.4 PROOF OF COROLLARY 1

This corollary directly follows from Theorem 2 as proposed by VSD (Wang et al., 2024b).

*Proof.* Theorem 2 by VSD establishes that the update rule for each particle $\theta_\tau$ at ODE time $\tau$ within a Wasserstein gradient flow is given by:

$$
\frac{d\theta_\tau}{d\tau} = \mathbb{E}_{t,\boldsymbol{\epsilon},c} \left[ \sigma_t \omega(t) \left( \nabla_{\boldsymbol{x}_t} \log p_t(\boldsymbol{x}_t|y^c) - \nabla_{\boldsymbol{x}_t} \log q_t^{\mu_\tau}(\boldsymbol{x}_t|c, y) \right) \frac{\partial \boldsymbol{g}(\theta_\tau, c)}{\partial \theta_\tau} \right].
\tag{29}
$$

In the case of rectified distribution, the update rule is modified as follows:

$$
\frac{d\theta_\tau}{d\tau} = \mathbb{E}_{t,\boldsymbol{\epsilon},c} \left[ \sigma_t \omega(t) \left( \nabla_{\boldsymbol{x}_t} \log \tilde{p}_t(\boldsymbol{x}_t|y) - \nabla_{\boldsymbol{x}_t} \log q_t^{\mu_\tau}(\boldsymbol{x}_t|c, y) \right) \frac{\partial \boldsymbol{g}(\theta_\tau, c)}{\partial \theta_\tau} \right],
\tag{30}
$$

which can be further simplified as:

$$
\begin{aligned}
\frac{d\theta_\tau}{d\tau} &= \mathbb{E}_{t,\boldsymbol{\epsilon},c} \left[ \sigma_t \omega(t) \left( \nabla_{\boldsymbol{x}_t} \log \left[ p(\boldsymbol{x}_t|y) r(\boldsymbol{x}_t|y) \right] - \nabla_{\boldsymbol{x}_t} \log q_t^{\mu_\tau}(\boldsymbol{x}_t|c, y) \right) \frac{\partial \boldsymbol{g}(\theta_\tau, c)}{\partial \theta_\tau} \right] \\
&= \mathbb{E}_{t,\boldsymbol{\epsilon},c} \left[ \sigma_t \omega(t) \left( \nabla_{\boldsymbol{x}_t} \log p(\boldsymbol{x}_t|y) - \nabla_{\boldsymbol{x}_t} \log q_t^{\mu_\tau}(\boldsymbol{x}_t|c, y) + \nabla_{\boldsymbol{x}_t} \log r(\boldsymbol{x}_t|y) \right) \frac{\partial \boldsymbol{g}(\theta_\tau, c)}{\partial \theta_\tau} \right] \\
&= \mathbb{E}_{t,\boldsymbol{\epsilon},c} \left[ \omega(t) \left( \boldsymbol{\epsilon}_\phi(\boldsymbol{x}_t, t, c, y) - \boldsymbol{\epsilon}_{\text{pretrain}}(\boldsymbol{x}_t, t, y) \right) \frac{\partial \boldsymbol{g}(\theta_\tau, c)}{\partial \theta_\tau} + \omega(t) \frac{\sigma_t}{\alpha_t} \nabla_{\theta_\tau} \log r(\boldsymbol{x}_t|y) \right].
\end{aligned}
\tag{31}
$$

Therefore, $\theta^{(i)}$ can be update by $\theta^{(i)} \leftarrow \theta^{(i)} - \eta \nabla_\theta \mathcal{L}_{\text{USD}}(\theta^{(i)})$, where:

$$
\begin{aligned}
\nabla_\theta \mathcal{L}_{\text{USD}} &= \nabla_\theta \mathcal{L}'_{\text{VSD}}(\theta) - \mathbb{E}_{t,\boldsymbol{\epsilon},c} \left[ \omega(t) \frac{\sigma_t}{\alpha_t} \nabla_\theta \log r(\boldsymbol{x}_t|y) \right], \\
\nabla_\theta \mathcal{L}'_{\text{VSD}} &= \mathbb{E}_{t,\boldsymbol{\epsilon},c} \left[ \omega(t) \left( \boldsymbol{\epsilon}_{\text{pretrain}}(\boldsymbol{x}_t, t, y) - \boldsymbol{\epsilon}_\phi(\boldsymbol{x}_t, t, c, y) \right) \frac{\partial \boldsymbol{g}(\theta, c)}{\partial \theta} \right].
\end{aligned}
\tag{32}
$$

Proof complete.

$\square$

# C   SUPPLEMENTARY EXPERIMENTS

This appendix contains supplementary experimental details. C.1 provides a detailed discussion of metric calculations. C.2 analyzes the influence of hyperparameters. C.3 visualizes the user-provided templates. C.4 demonstrates our method's scalability through cross-domain rectification. C.5 explores special cases to showcase practical applications. C.6 and C.7 present additional performance results. Note that the prompt list, comparisons, and results are left in Appendix G and Appendix H.

## C.1   METRICS

### C.1.1   FRÉCHET INCEPTION DISTANCE FOR GENERATION QUALITY

The Fréchet Inception Distance (Heusel et al., 2017) (FID) serves as our primary metric for evaluating generation quality by measuring the statistical distance between two image distributions. In our evaluation process, we compare our generated images against two different target distributions:

**Standard FID.**   To evaluate the quality gap between our generated 3D scenes and pretrained Stable Diffusion (Rombach et al., 2022) outputs, we establish a target distribution by sampling 60 images per prompt across 22 different prompts, yielding a total test set of 1,320 images. To mitigate pose bias in this distribution, we incorporate directional text descriptions such as "front view," "side view," and "back view" during sampling. However, we note that some pose bias remains, with frontal views being over-synthesized. For our generated distribution, we render 5 images from each 3D scene using uniformly sampled camera poses. The standard FID score is then calculated between this rendered set and our target distribution.

**Unbiased FID (uFID).**   To address the inherent pose bias present in standard FID evaluation, we develop an alternative metric called uFID. This approach begins with manual annotation of camera poses for all images in the standard test set. Using these annotations, we resample the test set to ensure equal representation across different poses. While this resampling strategy may result in some image duplication, it yields a more balanced distribution of viewpoints and textures. The uFID score is then computed between this pose-balanced dataset and our rendered images, providing a more equitable assessment of generation quality across different viewpoints.

### C.1.2   CATEGORIAL ENTROPY FOR GEOMETRIC CONSISTENCY

We evaluate the Multi-Face Janus problem using a pose classifier to analyze viewpoint consistency across different perspectives. The underlying principle is straightforward: in a scene with severe Multi-Face Janus issues, different viewpoints will yield similar classification probabilities because they share similar features. This similarity typically manifests as a strong bias toward a particular class (usually the canonical pose) in the average classification probability across viewpoints. Conversely, a geometrically consistent 3D scene will produce more diverse classification probabilities that average toward a more uniform distribution across viewpoints.

Based on this insight, we use the entropy of the average classification probability from different views as a metric for measuring the severity of multiplicity problems. Higher entropy values indicate greater diversity in information across viewpoints, while lower values suggest excessive pattern duplication in the generated scene. Formally, for each prompt, we calculate the entropy $R_{ent}$ as:

$$\bar{\boldsymbol{p}} = \frac{1}{n_v} \sum_{i=0}^{n_v} \boldsymbol{\Phi}(\boldsymbol{g}(\theta, c_i)),$$

$$R_{ent} = \frac{1}{n_{\bar{c}}} \sum_{i=0}^{n_{\bar{c}}} \bar{p}_i \log \bar{p}_i, \tag{33}$$

where $\boldsymbol{g}$ represents the renderer and $\boldsymbol{\Phi}$ the pose classifier. The probability vector $\bar{\boldsymbol{p}}$ consists of components $\bar{p}_i$. $n_v$ denotes the number of sampled views (default: 10), and $n_{\bar{c}}$ represents the number of pose categories. We implement this entropy evaluation using two different classification approaches:

**CLIP Entropy (cEnt).**   This method employs CLIP (Radford et al., 2021) as the classifier, using three textual descriptions that combine the original prompt with directional modifiers: "from

"A DSLR photo of a beagle in a detective's outfit."

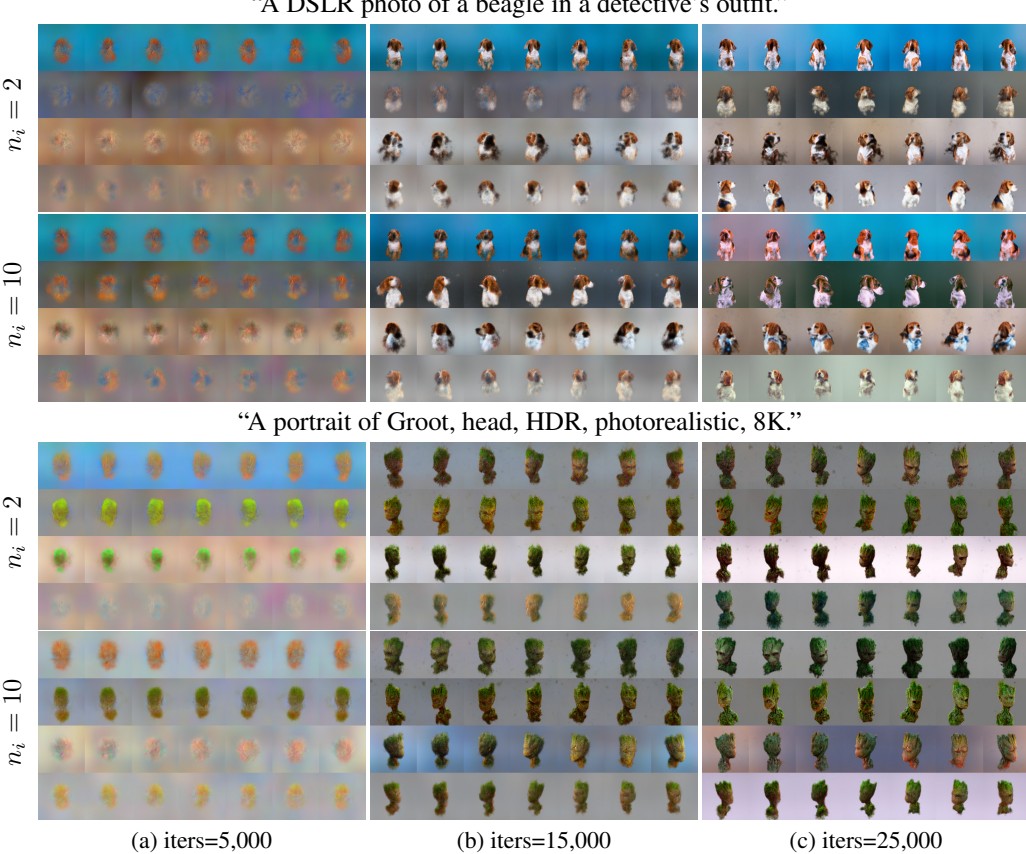

"A portrait of Groot, head, HDR, photorealistic, 8K."

(a) iters=5,000  (b) iters=15,000  (c) iters=25,000

Figure 6: Influence of number of BNF intervals $n_i$ for multiple particles. Experiments compare performance with $n_i = 2$ versus $n_i = 10$ intervals. The finer granularity of BNF intervals ($n_i = 10$) leads to better synchronization between particle generations, mitigating the Multi-Face Janus Problem.

front view", "from side view", and "from back view". These descriptions establish three distinct categories for classifying input images.

**Pose Entropy (pEnt).** This variant utilizes our specially designed pose classifier for categorization, providing a more direct assessment of pose-related geometric consistency.

### C.1.3 CLIP Score for Textual Alignment.

To evaluate textual alignment, we calculate the CLIP score by measuring the negative cosine similarity between CLIP feature embeddings of the rendered images and their corresponding text prompts, following Wang et al. (2024a).

### C.2 Hyperparameter Analysis

**Influence of BNF interval $n_i$.** The BNF time scheduler controls the sampling intervals for score distillation. A larger $n_i$ provides finer control over the sampling process, making it more closely resemble the DDPM sampling process. Through our single-particle optimization experiments, we observe that adjusting $n_i$ values demonstrates minimal influence on training dynamics until the terminal phase of optimization. During this final stage, larger $n_i$ configurations become particularly impactful. For example, when the sampling time step is limited to the interval $[100, 0]$, the model tends to overfit, often producing oversaturated colors. We typically use early stopping to prevent this. However, in simultaneous multi-particle optimization, we observe that a larger BNF $n_i$ improves training quality. As shown in Fig. 6, the training process for different BNF values demonstrates

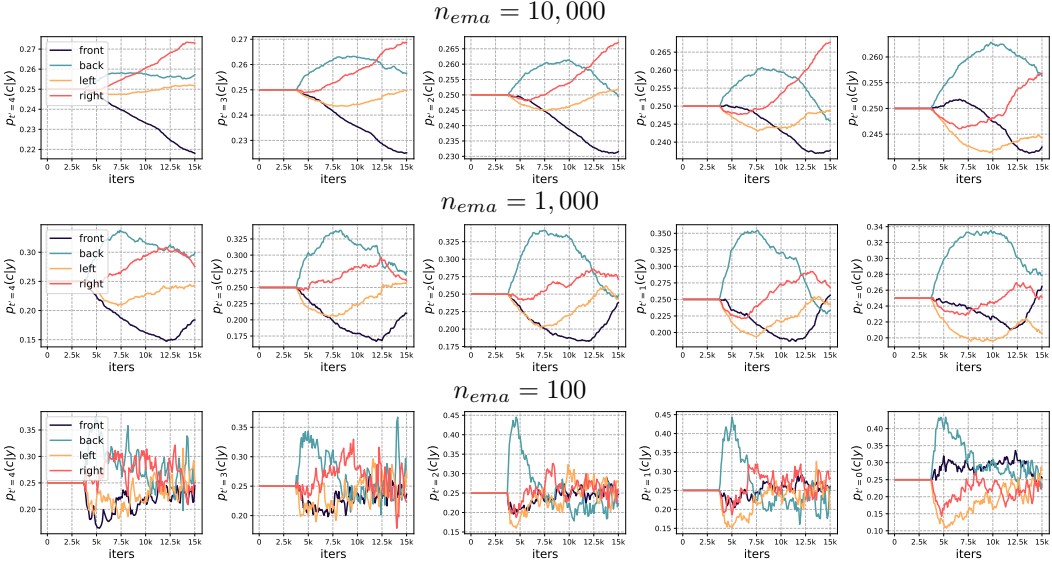

Figure 7: The impact of varying the number of valid EMA steps $n_{ema}$ on pose probability distributions over time. Each curve shows $\bar{p}_t(\bar{c}|y)$, representing how pose distributions evolve during training. For visualization clarity, we map the continuous time $t$ to discrete indices $t'$, where each index spans 100 timesteps (e.g., $t' = 0$ corresponds to $t \in [0, 100]$). A lower $n_{ema}$ enables timely correction of distribution bias, resulting in more stable probability distributions (*i.e.*, please zoom in for a better view of the y-axis).

"A kangaroo wearing boxing gloves."

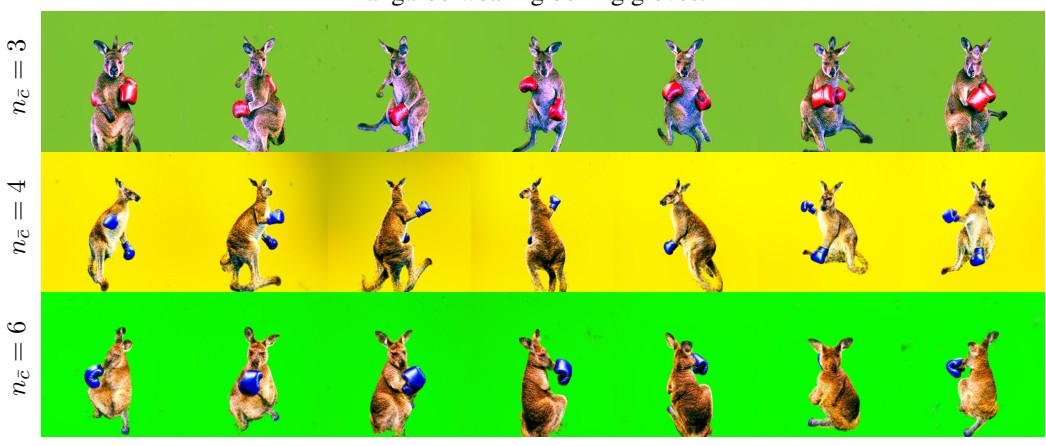

Figure 8: Influence of the number of templates $n_{\bar{c}}$. Varying $n_{\bar{c}}$ shows minimal impact on performance, as our pose classifier is designed for coarse categorization and cannot effectively distinguish fine-grained poses at higher $n_{\bar{c}}$ values.

this effect. When $n_i = 2$, the training is imbalanced across particles. For example, in the case of "beagle," the first particle learns the information more quickly, while the fourth particle of "Groot" progresses more slowly. This imbalance causes the fastest particle to converge to one mode of the distribution, such as all back views for "beagle", while other particles converge to different modes (e.g., front view, back view, etc.). This results in an undesirable outcome, where the overall distribution across particles appears uniform, but each individual particle suffers from the Multi-Face Janus problem—one is biased toward front views, while another is biased toward back views, which contradicts our goal. To resolve this issue, we ensure that all particles converge at a consistent rate. Our BNF time scheduling controls the interval of sampling time steps, where a larger BNF value ($n_i$) results in time steps being sampled within a narrower range during the early training phase. This promotes a more uniform optimization process across particles. As a result, training consistency

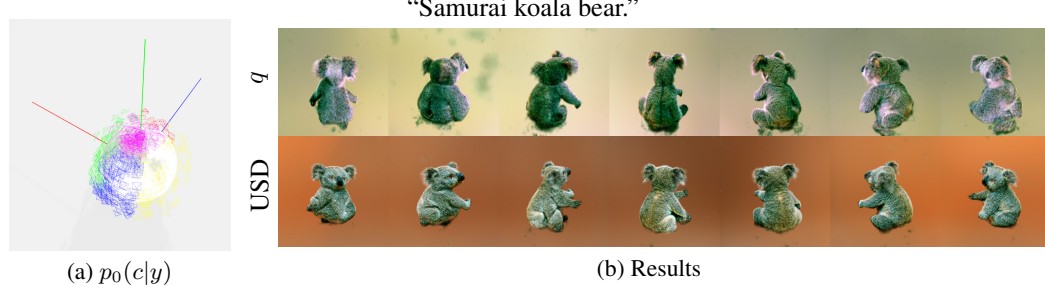

(a) $p_0(c|y)$                          (b) Results

Figure 9: Comparison with the $q$ sampling and our USD. $q$ sampling modifies uniform view sampling using estimated pose probabilities $p_0(c|y)$. (a) shows the estimated $p_0(c|y)$ distribution for the prompt, with probabilities approximately $[0.1, 0.6, 0.15, 0.15]$ for the front (red), back (blue), left (yellow), and right (green) views. (b) shows the corresponding generation results.

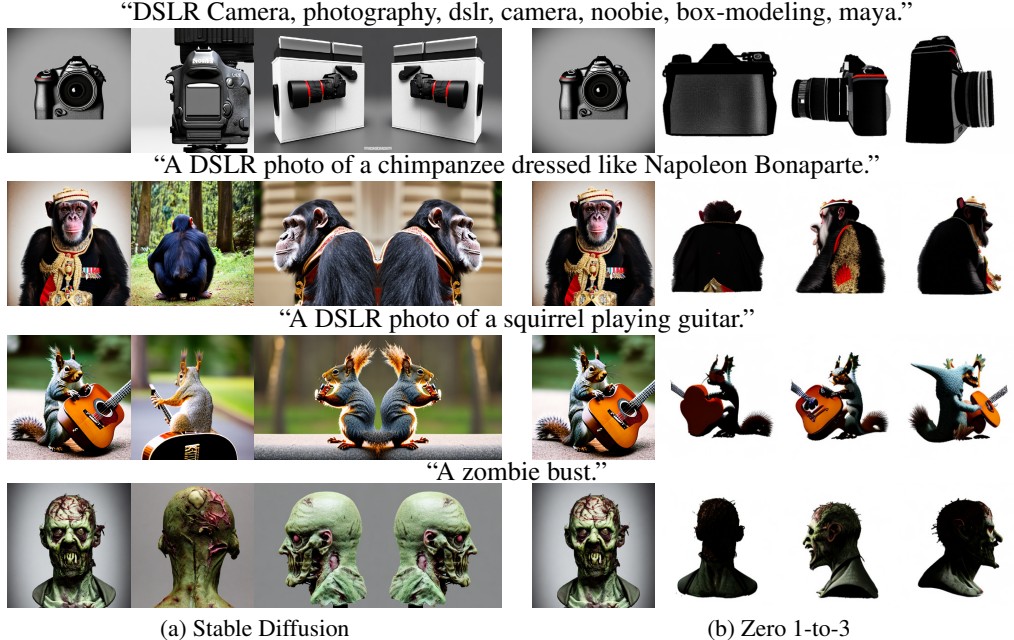

(a) Stable Diffusion                          (b) Zero 1-to-3

Figure 10: Examples of template images. (a) Templates manually curated from Stable Diffusion (Rombach et al., 2022) generations. (b) Multi-view images obtained using Zero 1-to-3 (Liu et al., 2023). Note that the multi-view image need not be completely 3D consistent and high-quality.

improves, reducing the risk of biased convergence. In Fig. 6, we show the generation effect with $n_i = 10$, which better aligns the modified distribution across all four particles.

**Influence of valid EMA steps $n_{ema}$.** The valid number of EMA steps, $n_{ema}$, ensures that the weights of the last $n_{ema}$ steps account for more than 90% of the total, while weights beyond this threshold are negligible and can be approximated as invalid. We set $n_{ema}$ to 100, 1000, and 10000, respectively, to simulate distribution updates at different speeds. We plot the probability curve $\bar{p}_t(\bar{c}|y)$ during the first five EMA intervals (i.e., $t \in [0, 500]$ steps for original sampling, with the first half held at 0 because the BNF scheduler has not yet sampled the current interval). The results in Fig. 7 show that an overly long EMA time step prevents the model from capturing the real-time pose distribution, leading to an inability to correct distribution bias. In practice, we typically choose $n_{ema} = 100$ to ensure effective estimation of the distribution.

**Influence of number of views $n_{\bar{c}}$.** We study how varying the number of pose categories ($n_{\bar{c}} = 3$, 4, and 6) affects overall performance. Our experiments, as illustrated in Fig. 8, show that increasing the number of categories from 4 to 6 produces minimal improvement in 3D consistency. This

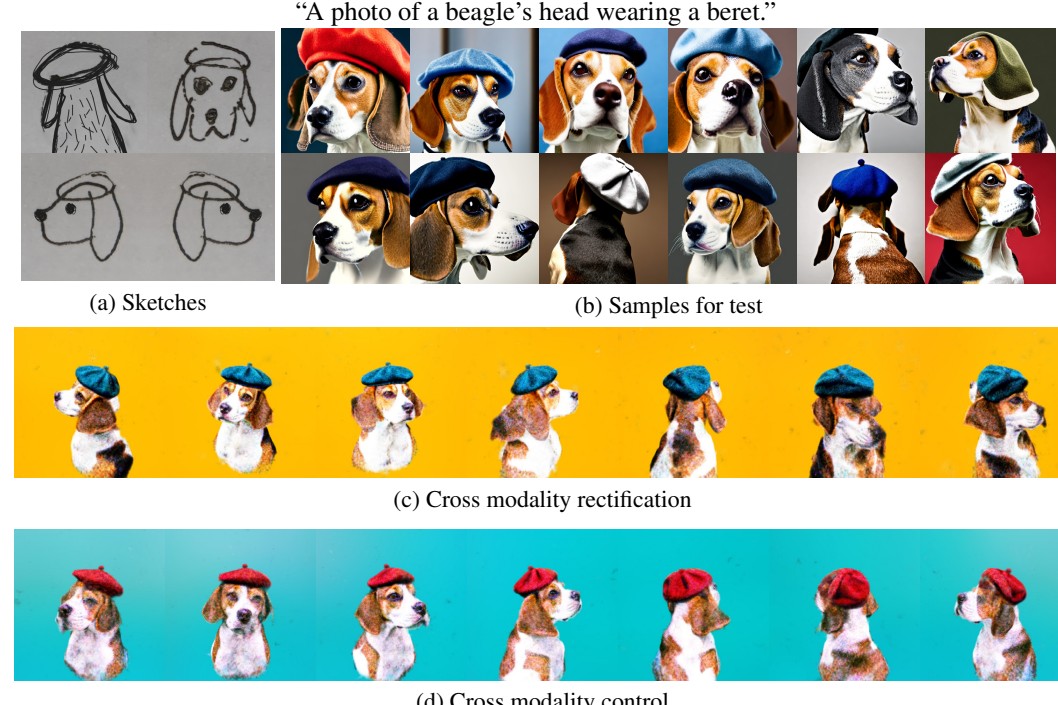

"A photo of a beagle's head wearing a beret."

(a) Sketches                    (b) Samples for test

(c) Cross modality rectification

(d) Cross modality control

Figure 11: Cross-modality generation results. (a) Amateur sketches drawn by the authors. (b) Initial Stable Diffusion generations for evaluation (quantitative results in Table 2). (c) Results with sketch-based rectification. (d) Generations with pose-controlled sketch guidance, demonstrating significant quality improvements.

performance plateau stems from our pose classifier's design, which lacks sensitivity to fine-grained pose distinctions, creating a natural ceiling when presented with more detailed pose categories.

**Sampling** $q$. We examine an alternative approach for sampling the target distribution $q$ using estimated probability $p_t(c|y)$, similar to the first-stage methodology in DreamControl (Huang et al., 2024). As shown in Fig. 9, this sampling strategy overemphasizes densely populated regions of the pose space while providing insufficient supervision for less frequent viewpoints. This imbalance leads to compromised geometric consistency in the generated results.

### C.3 TEMPLATES FOR POSE CLASSIFIER

We generate reference template images using Stable Diffusion (Rombach et al., 2022) by combining the original prompt with directional text modifiers: "from front view," "from side view," and "from back view." With the set of generated images, users can simply select some with different poses as the templates. Additionally, users also have the option to generate templates using Zero-1-to-3 (Liu et al., 2023).

As shown in Fig. 10, the template images do not need to be 3D consistent. They only need to convey basic pose information. Our experiments in Appendix C.4 demonstrate this flexibility through a cross-modal experiment where we successfully use simple sketches as templates.

### C.4 CROSS MODALITY RECTIFICATION

We demonstrate our pose estimator's scalability through a cross-domain experiment using hand-drawn sketches. For this experiment, we draw four sketches (see Fig. 11(a)) corresponding to the prompt and construct a pose classifier based on these sketches. When applied to real images, this sketch-based classifier accurately computes both masks and probabilities. In Fig. 11(b), we sample some images for testing the performance of the sketch-based classifier. Quantitative results are presented in Table 2.

Table 2: Cross-modal classifier predictions for images in Fig. 11(b). Images are arranged in two rows: IDs 1-6 (top) and IDs 7-12 (bottom). Cell colors indicate manually annotated ground-truth pose categories. Note: Ambiguous images (*e.g.*, ID 1) that could be classified as multiple categories (*e.g.*, "front" or "left") are highlighted with lighter color. **Bold** values indicate maximum classifier probabilities.

| ID | Back | Front | Left | Right | ID | Back | Front | Left | Right |
|----|------|-------|------|-------|----|------|-------|------|-------|
| 1 | 0.0127 | **0.7895** | 0.1773 | 0.0204 | 7 | 0.0017 | 0.4411 | **0.5447** | 0.0123 |
| 2 | 0.0037 | 0.2843 | 0.0217 | **0.6902** | 8 | 0.0100 | 0.1306 | 0.0069 | **0.8522** |
| 3 | 0.0013 | **0.9684** | 0.0152 | 0.0149 | 9 | **0.8361** | 0.0154 | 0.0792 | 0.0691 |
| 4 | 0.0016 | **0.8374** | 0.1291 | 0.0318 | 10 | 0.0006 | 0.1418 | **0.8526** | 0.0048 |
| 5 | 0.0781 | 0.0637 | 0.0004 | **0.8575** | 11 | **0.8979** | 0.0610 | 0.0236 | 0.0173 |
| 6 | 0.0018 | 0.1153 | **0.8795** | 0.0033 | 12 | 0.0393 | 0.0368 | 0.0055 | **0.9182** |

"Samurai koala bear."

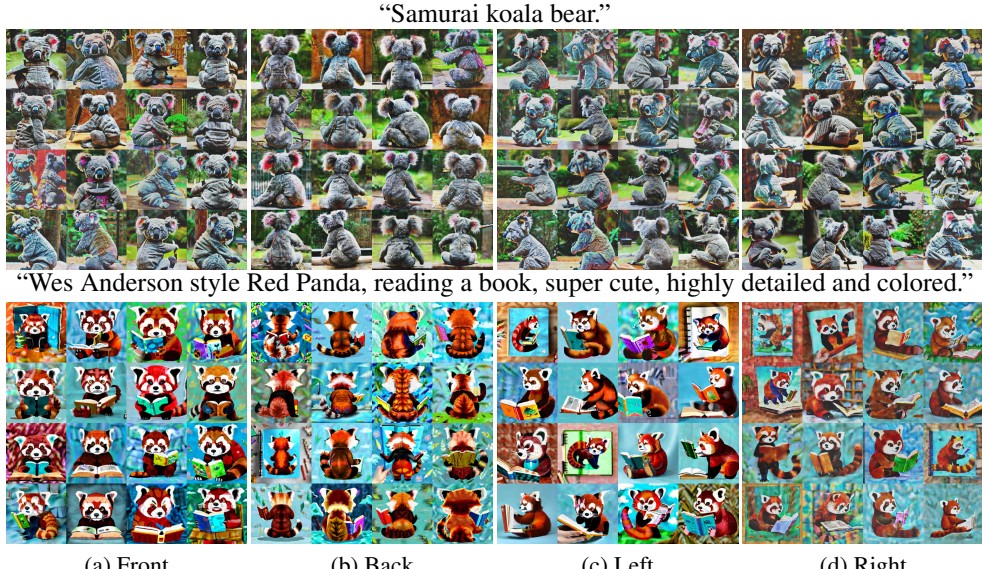

"Wes Anderson style Red Panda, reading a book, super cute, highly detailed and colored."

(a) Front          (b) Back          (c) Left          (d) Right

Figure 12: Pose-controlled generation using classifier outputs. Each batch demonstrates generation results using one of the four classifier logits, corresponding to "front", "back", "left", and "right" poses, respectively.

Fig. 11(c) and (d) present the 3D generation results. (c) shows scenes generated using manual sketch guidance, while (d) demonstrates results using pose control supervision (detailed in Appendix C.5). The latter approach notably enhances the 3D consistency of the output.

## C.5 CONTROLLABILITY AS A SPECIAL CASE OF USD

We further explore conditional control building upon our previous classifier. While our experimental setup considered a uniform distribution of poses across all angles, real-world applications may not require such comprehensive coverage. The number of discrete poses can be reduced when we can specify a more precise range of angles.

In cases where the perspective can be constrained to a narrow range (*e.g.*, front views from -45 to 45 degrees) that can be summarized with only one template image, the rectifier function can be simplified to $r_\xi^{ctrl}(\boldsymbol{x}_t|y) = p_\xi(\bar{c}|\boldsymbol{x}_t, y)$. This simplification is similar to approaches used in PnP (Graikos et al., 2022) and DPS (Chung et al., 2022), where irrelevant $p_t(\bar{c}|y)$ terms are eliminated. Under these conditions, the generation process is controlled by a single pose, effectively targeting just one specific conditional sub-distribution.

For implementation, we can optimize the classifier using a single logit to achieve this control. While this approach resembles PnP (Graikos et al., 2022), our method enables multi-particle optimization without requiring specialized classifiers for noisy images, which turns out to be more flexible.

Table 3: Runtime comparisons. All measurements are reported in minutes.

| USD | VSD | SDS | Debiased-SDS | PerpNeg | ESD | SDS-Bridge |
|------|--------|-------|--------------|---------|--------|------------|
| 297.34 | 180.70 | 43.35 | 43.58 | 45.97 | 201.55 | 73.24 |

Table 4: User study on geometric consistency. Experts ranked the generated results from different baseline methods based on geometric consistency. Rankings are converted to scores based on position, with higher-ranked options receiving higher scores.

| USD | VSD | SDS | Debiased-SDS | PerpNeg | ESD | SDS-Bridge |
|------|------|------|--------------|---------|------|------------|
| 4.80 | 3.27 | 2.81 | 2.44 | 1.96 | 3.21 | 2.50 |

Table 5: Success rates for Janus-free generation. Scoring system assigns penalties of 0.5 for local feature duplication and 1.0 for global feature duplication. While our method effectively mitigates global feature duplication, local duplications occasionally persist due to inherent limitations of score distillation algorithm.

| Prompts | Mean | Std | Median | Mode | Min | Max |
|---------|------|--------|--------|-------|-----|-----|
| "kangaroo" | 0.65 | 0.5296 | 0.5 | 0.5 | 0 | 1.5 |
| "bear" | 0.94 | 0.5270 | 0.75 | 0.5/1 | 0.5 | 2 |

As demonstrated in Fig. 12, our approach achieves precise pose control in 2D particle generation. The examples show more efficient control compared to using textual descriptions. We've successfully extended this functionality to both cross-modal (sketch-guided) and 3D-prior based generation, enhancing geometric consistency as shown in Fig. 11(d) and Fig. 16(b). Furthermore, by leveraging DINOv2 (Oquab et al., 2023)'s cross-modal matching capabilities, we can explore novel applications. For instance, using airplane images to guide the generation of flying eagles. We believe this opens up promising avenues for expanding the practical applications of our theoretical framework.

## C.6 Runtime Evaluation

We conduct our experiments at $256 \times 256$ resolution using a single Nvidia GeForce RTX 4090 GPU. While USD and VSD share the same framework, other comparison methods are implemented using threestudio (Guo et al., 2023). To ensure a fair comparison, all methods are evaluated using their default settings to achieve convergence. As shown in Table 3, USD requires longer computation times compared to baseline methods, primarily due to the additional back-propagation through the diffusion U-Net (Ronneberger et al., 2015) required by the rectifier function.

Despite the increased computational overhead, our method's substantial improvements in geometric consistency justify this trade-off. The computational cost remains practical for pose control applications (in Appendix C.5), as pose control is typically only necessary during the initial stages of shape formation. Future research directions could focus on developing optimization strategies to enhance computational efficiency while maintaining the method's performance advantages.

## C.7 User Study and Discussions on the Success Rate

We conduct a user study involving more than 25 human experts to evaluate and rank different methods based on their overall geometric consistency. The average ranks for all methods are reported in Table 4.

Additionally, we analyze the success rate of generating Janus-free models. Given the inherent randomness of score distillation methods, achieving perfect geometric consistency remains challenging. To quantify inconsistencies, we develop a systematic rating system. The system evaluates both global and local features using the formula $R_{score} = (n_{cnt}^g - n_{gt}^g) + 0.5 \times (n_{cnt}^l - n_{gt}^l)$, where for global features (such as faces or body), $n_{cnt}^g - n_{gt}^g$ represents the difference between the actual count and the expected count, while for local features (such as legs, arms, tails), each duplicated feature contributes 0.5 to the score. Here, $n_{cnt}^l$ and $n_{gt}^l$ denote the actual and expected counts of local features respectively. Note that the expected count is the correct number that the scene should occur. For instance, in the case of generating a bust, the face's expected count is 1. Table 5 presents

Table 6: Classification performance and ablation studies. We compare three variants: orientation classifier without texture score, texture classifier without orientation score, and the complete model using both scores.

| Metric | Full | Orient | Texture |
|---|---|---|---|
| Average Accuracy | 0.7846 | 0.6328 | 0.7699 |
| Average Precision | 0.7986 | 0.6718 | 0.8013 |
| Average Recall | 0.7426 | 0.5934 | 0.7396 |
| Average F1 Score | 0.7439 | 0.5942 | 0.7308 |

"Samurai koala bear."                     "A kangaroo wearing boxing gloves."

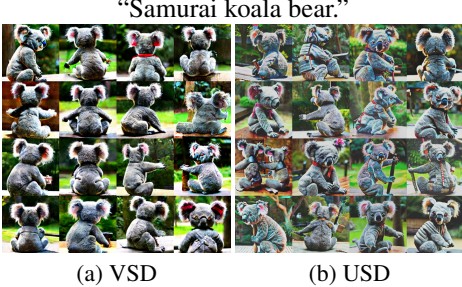
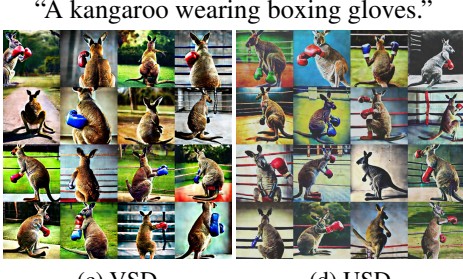

(a) VSD    (b) USD        (c) VSD    (d) USD

Figure 13: 2D score distillation comparing VSD (Wang et al., 2024b) and USD. The prompts are augmented with auxiliary view descriptions ("from side view, from back view") to capture multi-perspective information. Due to the original distribution's bias toward back-view angles, VSD generates predominantly back-view results, while USD successfully rectifies this distributional bias to produce more balanced viewpoints.

the Multi-Face Janus scores for two test prompts, demonstrating our method's performance. We provide a detailed discussion of potential solutions in Appendix F.

## D    CLASSIFIER EXPERIMENTS

In this appendix, we evaluate our pose classifier's performance using the annotations described in Sec. 4.1. We assess both the texture and orientation branches independently. Table 6 presents the classification performance of the main classifier and its two variants. This ablation study validates our chosen classifier architecture.

## E    VALIDATION EXPERIMENTS ON 2D SAMPLING

We validate the performance of uniform score distillation using a set of 2D particles. Starting with 16 initialized particles, we conduct training over 4,000 iterations. The comparison between USD and variational score distillation is illustrated in Fig. 13. Our results demonstrate that USD successfully achieves a more balanced distribution compared to the biased pre-trained distribution.

Fig. 14 shows the pose distribution statistics across 10 intervals for both USD and VSD. Here, $\bar{p}_t(\bar{c}|y)$, which represents the expectation of $\bar{p}_t(\bar{c}|\boldsymbol{x}, y)$ over $\boldsymbol{x}_t$, indicates the current pose distribution and reveals training bias progression. While VSD exhibits a strong bias toward specific distributions during training (due to the usage of auxiliary prompts), our method maintains an approximately uniform distribution throughout the process.

## F    DISCUSSION ON LIMITATIONS AND FUTURE WORKS

This discussion examines our work's boundaries while identifying promising paths for subsequent research. We identify several key limitations and opportunities for advancement.

A primary limitation of this work is generation speed. The bottleneck lies in the U-Net (Ronneberger et al., 2015) gradient back-propagation introduced by the rectifier function, which requires further

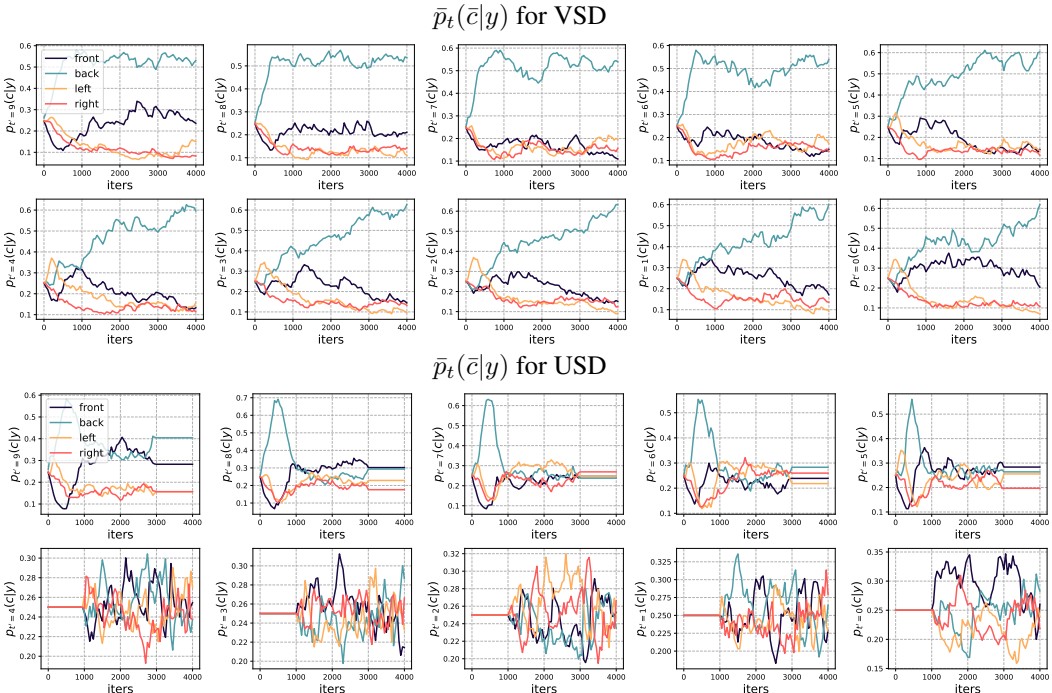

Figure 14: Comparison of pose probability distributions $\bar{p}_t(\bar{c}|y)$ between VSD (Wang et al., 2024b) and USD across different timestep intervals $t$. While VSD converges to a biased distribution, USD maintains an approximately uniform distribution across camera poses.

"A person's face."

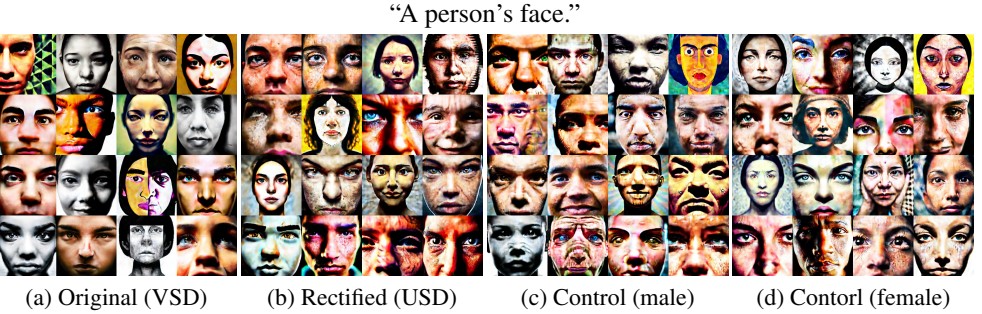

(a) Original (VSD)    (b) Rectified (USD)    (c) Control (male)    (d) Contorl (female)

Figure 15: Addressing semantic distributional bias using a CLIP (Radford et al., 2021) classifier. (a) Results from VSD (Wang et al., 2024b) exhibit inherent gender bias, predominantly generating female subjects. (b) By incorporating a CLIP-based male/female classifier, our method achieves balanced gender distribution. (c) and (d) demonstrate fine-grained control over specific gender attributes, enabling targeted generation of male and female subjects respectively.

optimization. Future research could explore methods to effectively bypass U-Net gradient backpropagation or develop a score-free optimization framework similar to MicroDreamer (Chen et al., 2024).

Another significant challenge concerns 3D consistency of localized features. While USD eliminates bias in the overall data distribution, its reliance on the score distillation algorithm, which lacks explicit geometric consistency supervision, can lead to geometrically inconsistent content, potentially limiting practical applications. Addressing this limitation requires incorporating multi-perspective supervision during generation. Notably, the special case discussed in Appendix C.5 demonstrates a potential supervision mechanism for score distillation that warrants further investigation.

Beyond current technical limitations, we propose new directions for control-based synthesis that expand on cross-modal approaches (see Appendix C.4). For instance, Fig. 16 demonstrates an ex-

"A platypus, dressed in a video game pixelated costume, steps on a pixelated surfboard and holds a squid weapon that emits 8-bit light effects."

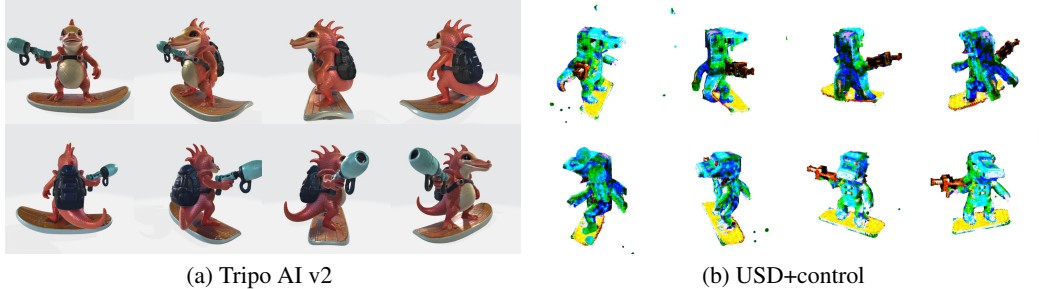

(a) Tripo AI v2                                       (b) USD+control

Figure 16: Demonstration of the integration of USD with 3D-based methods (Tripo AI).

periment using an imaginative prompt. As shown in Fig. 16(a), the 3D generative model Tripo AI v2* captures basic geometric elements effectively, but still faces challenges when interpreting more abstract or imaginative descriptions (*i.e.*, "pixelated costume" and "pixelated surfboard") due to its 3D modeling constraints. In contrast, our approach leverages selected Tripo AI renderings for pose control (Fig. 16(a)), resulting in a more accurate prancing effect that better matches the text description, as demonstrated in Fig. 16(b). While our model is trained from scratch and may lack geometric refinement, fine-tuning it from a geometrically consistent base model (Zheng et al., 2024) can yield results that excel in both geometric accuracy and textural detail.

Finally, highlighting the versatility of our approach, our USD algorithm demonstrates considerable extensibility beyond pose classification and 3D generation. Fig. 15 showcases its application in addressing gender distribution bias in image generation using the CLIP (Radford et al., 2021) classifier, enabling independent control over gender representation. This adaptability suggests that USD could be applied to address other forms of algorithmic bias with different classifier architectures.

## G  PROMPTS

In our experiments, we use the prompts introduced in the existing works such as SDS (Poole et al., 2022) and VSD (Wang et al., 2024b) as shown in Table 7.

Table 7: Experimental prompt list. Each prompt is augmented with auxiliary view descriptors "from side view, from back view".

| ID | Prompt Description | ID | Prompt Description |
|---|---|---|---|
| 1 | An airplane made out of wood. | 12 | A peacock on a surfboard. |
| 2 | A bald eagle carved out of wood features. | 13 | A portrait of Groot, head, HDR, photorealistic, 8K. |
| 3 | A blue motorcycle. | 14 | A sea turtle. |
| 4 | A dragon-shaped teapot. | 15 | A zombie bust. |
| 5 | A DSLR photo of a beagle in a detective's outfit. | 16 | A 3D printed white bust of a man with curly hair. |
| 6 | A DSLR photo of a chimpanzee dressed like Napoleon Bonaparte. | 17 | DSLR Camera, photography, dslr, camera, noobie, box-modeling, maya. |
| 7 | A DSLR photo of a football helmet. | 18 | Mecha vampire girl chibi. |
| 8 | A kingfisher bird. | 19 | Robot with pumpkin head. |
| 9 | A fantasy painting of a dragoncat. | 20 | Robotic bee, high detail. |
| 10 | A kangaroo wearing boxing gloves. | 21 | Samurai koala bear. |
| 11 | A DSLR photo of a squirrel playing guitar. | 22 | Wes Anderson style Red Panda, reading a book, super cute, highly detailed and colored. |

_______________

*https://lumalabs.ai/genie

# H  ADDITIONAL COMPARISON AND RESULTS

In this part, we present additional comparisons and results.

"A DSLR photo of a beagle in a detective's outfit."

"A portrait of Groot, head, HDR, photorealistic, 8K."

"A kangaroo wearing boxing gloves."

Figure 17: Additional Comparisons with ESD and SDS-Bridge.

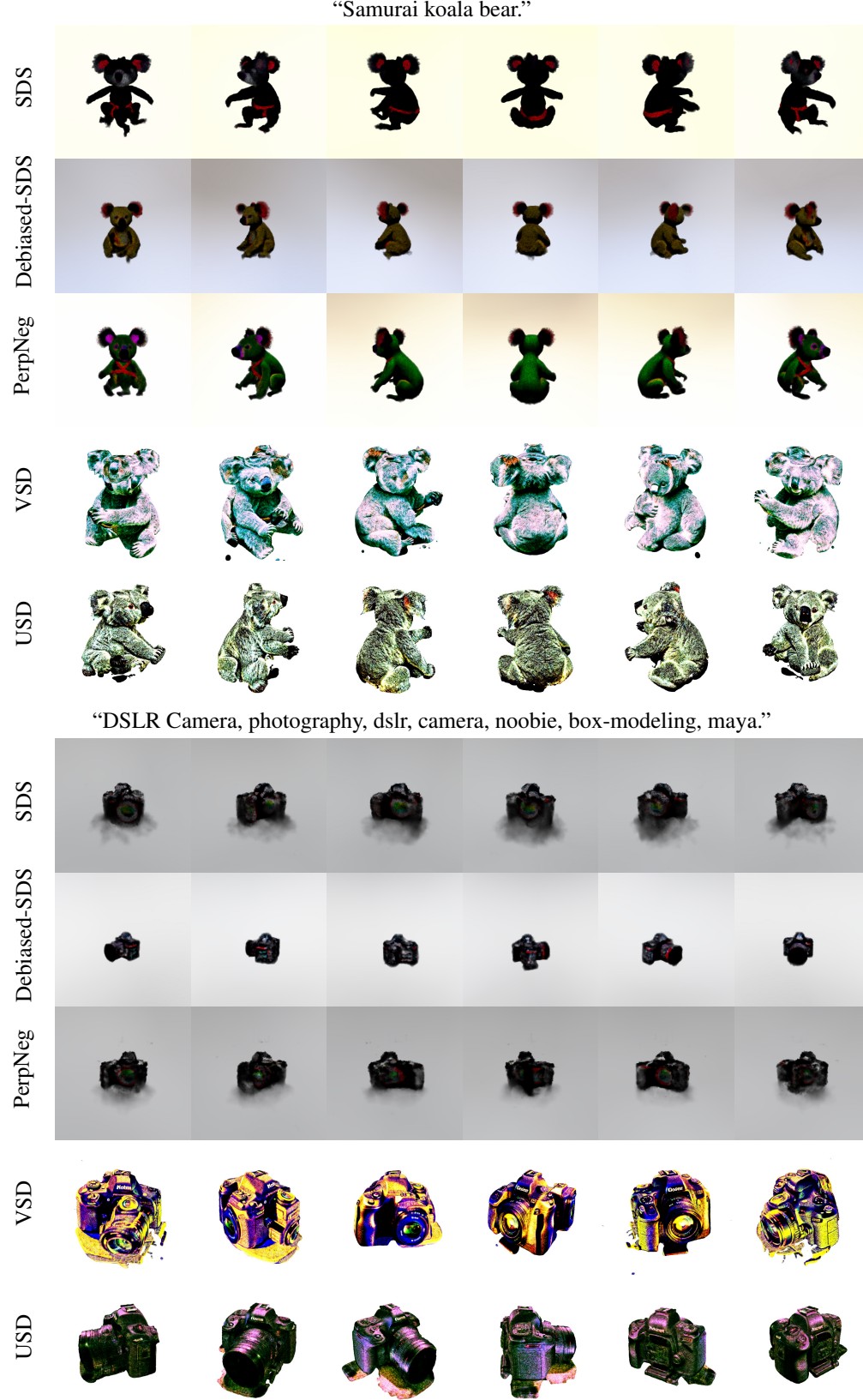

Figure 18: Additional Comparisons.

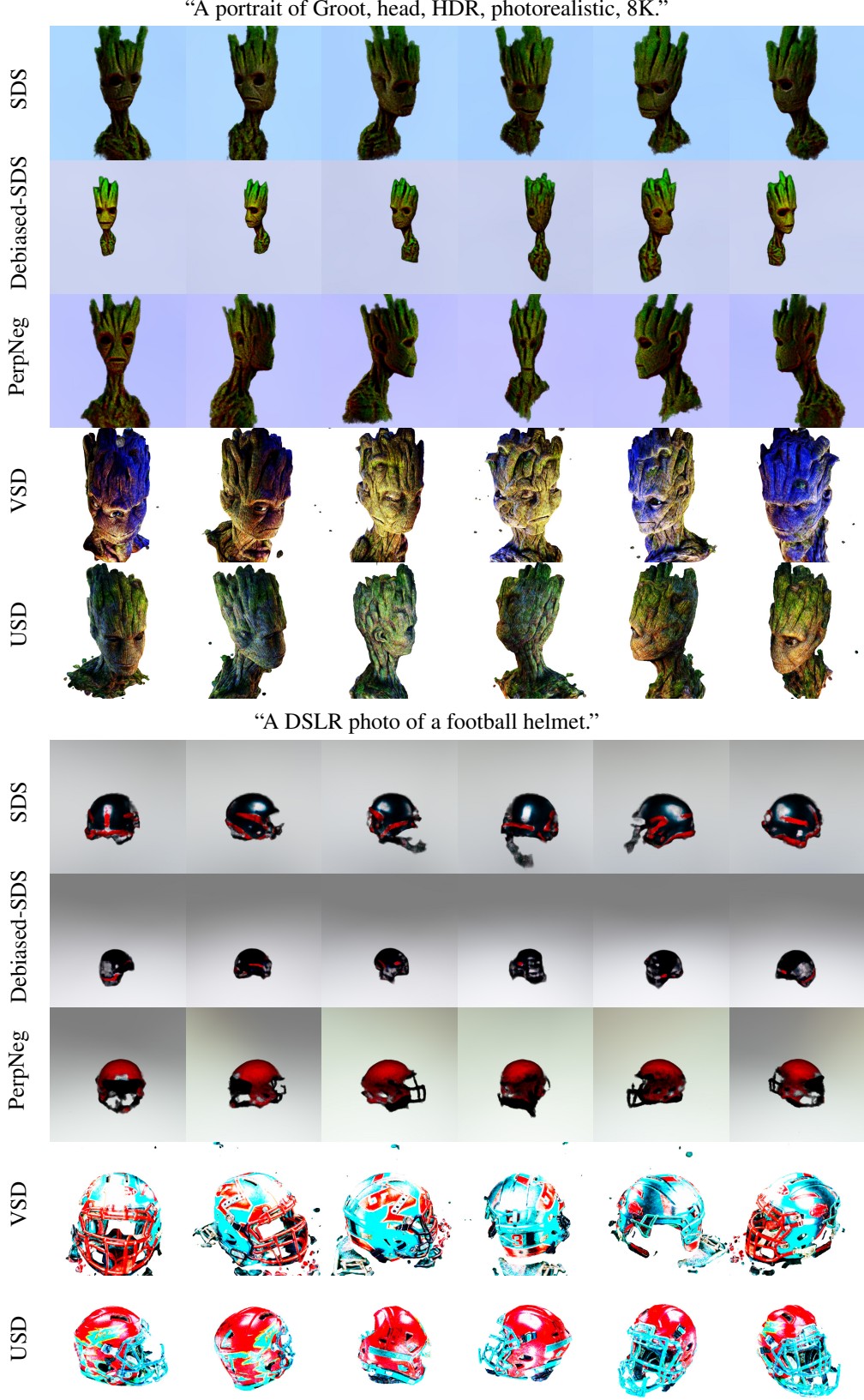

Figure 19: Additional Comparisons.

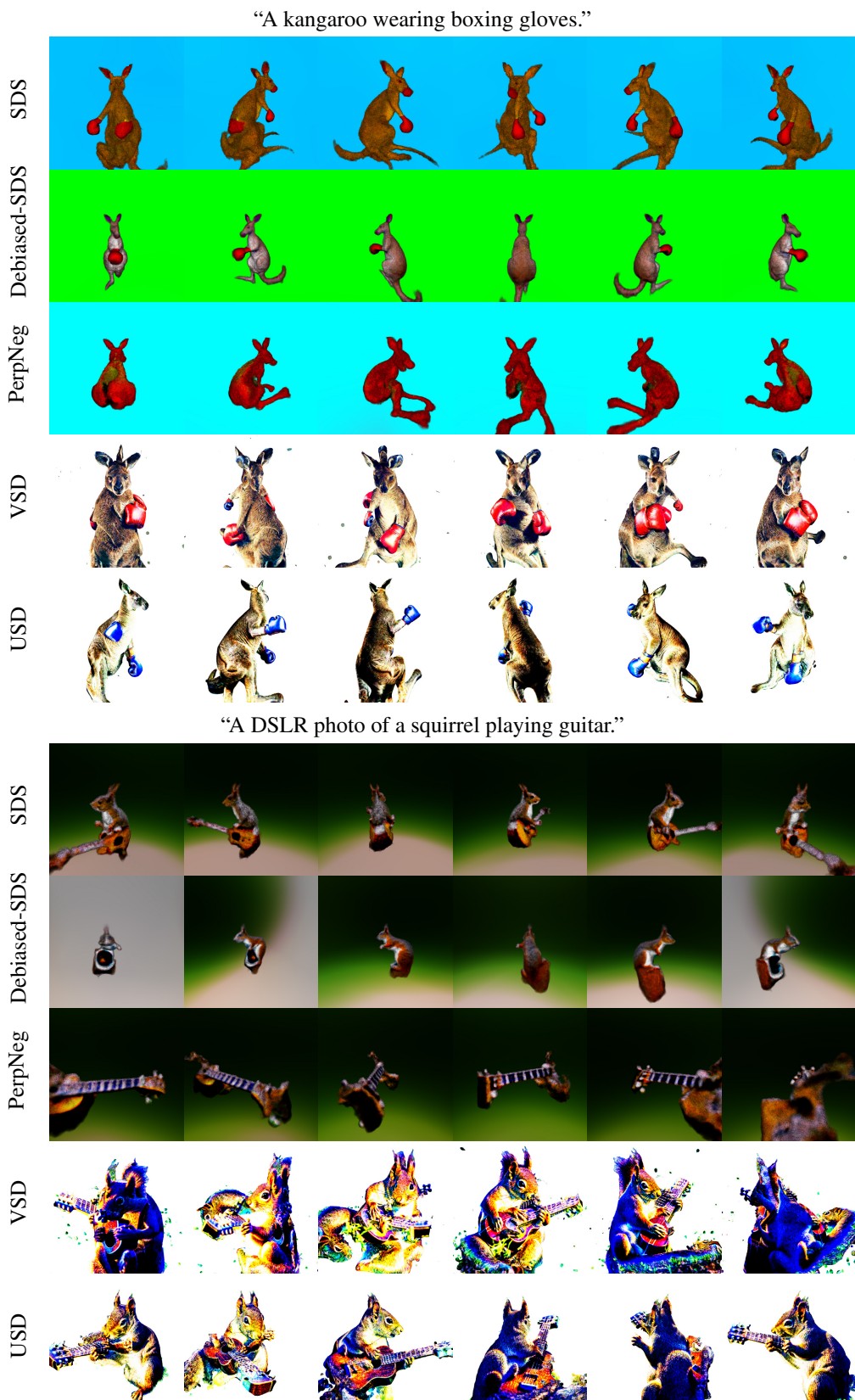

Figure 20: Additional Comparisons.

"A DSLR photo of a beagle in a detective's outfit."

"Mecha vampire girl chibi."

"A peacock on a surfboard."

"A blue motorcycle."

"An airplane made out of wood."

"Robotic bee, high detail."

"A DSLR photo of a chimpanzee dressed like Napoleon Bonaparte."

Figure 21: More examples.

"A bald eagle carved out of wood features."

"A kingfisher bird."

"Robot with pumpkin head."

"A dragon-shaped teapot."

"A sea turtle."

"A zombie bust."

"A fantasy painting of a dragoncat."

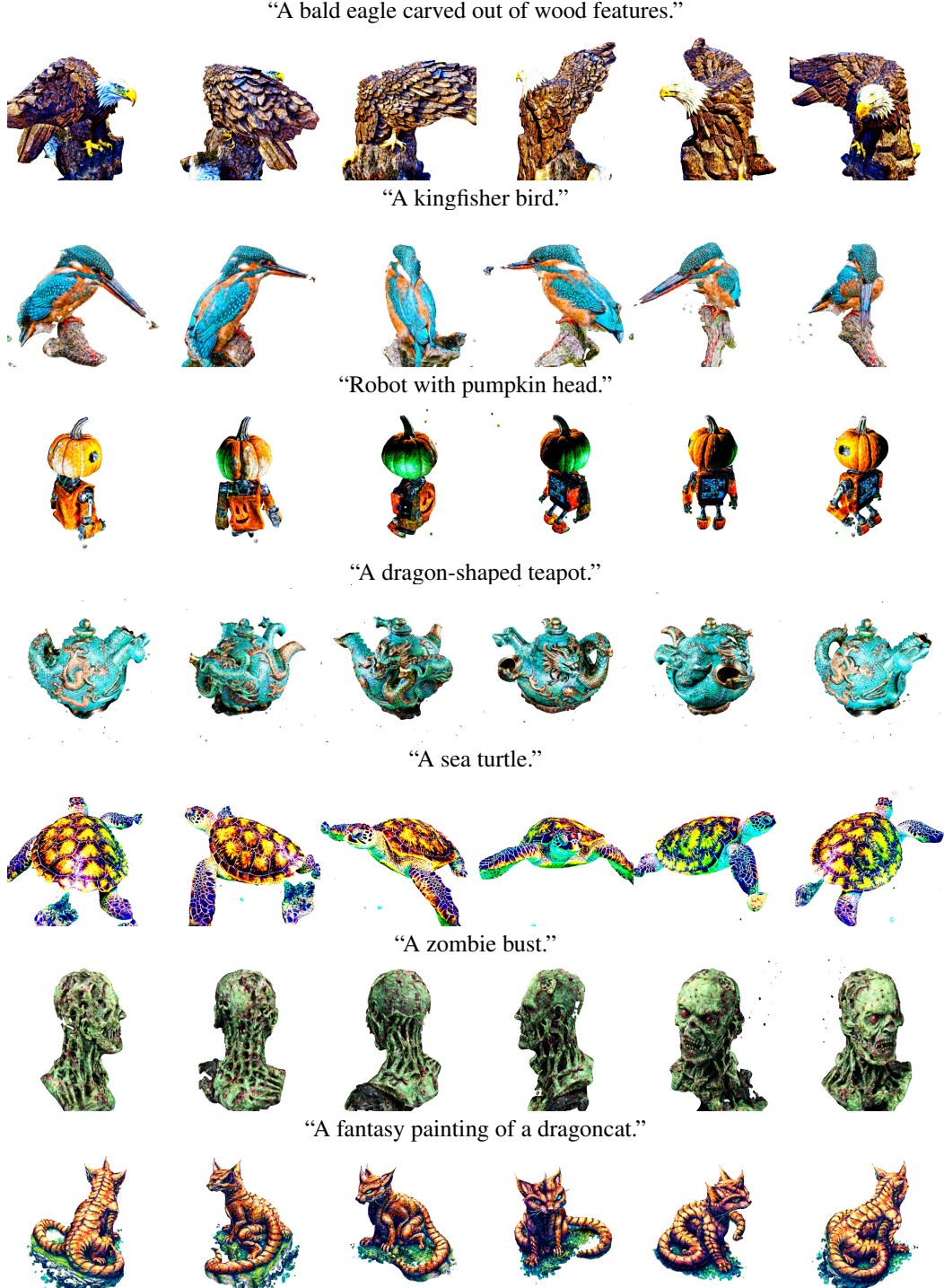

Figure 22: More examples.

