# OpenReview forum: "RecDreamer: Consistent Text-to-3D Generation via Uniform Score Distillation"
_ICLR.cc/2025/Conference — ICLR 2025 Poster_

### Official Review · Reviewer_X3Gz · 2024-10-31

**Soundness:** 2
**Presentation:** 3
**Contribution:** 2
**Rating:** 6
**Confidence:** 5

**Summary:**

The paper presents RecDreamer, an approach to address the Multi-Face Janus problem in text-to-3D generation, which arises from geometric inconsistencies across different poses of 3D assets. The authors propose a technique called uniform score distillation, which modifies the underlying data distribution to ensure pose variation follows a uniform distribution. This is achieved by a rectification process that adjusts the density distribution, facilitated by a training-free classifier that estimates pose categories. The approach aims to eliminate biases towards a canonical pose and improve geometric consistency without compromising rendering quality. The experimental results demonstrate the effectiveness of RecDreamer in achieving consistent 3D asset generation across various poses.

**Strengths:**

* An approach to address data bias issue: The paper introduces a solution to a well-known problem in text-to-3D generation by addressing the biases in data distribution through uniform score distillation.
The use of a training-free classifier to estimate pose categories is an efficient and novel aspect of the approach, avoiding the need for additional training.

* Comprehensive Methodology:
The integration of reverse Kullback-Leibler divergence in the score distillation framework is well-articulated, allowing for the seamless incorporation of the rectified distribution.
The method's ability to maintain rendering quality while resolving the Multi-Face Janus problem is a significant advantage.

* Experimental Validation:
The paper provides thorough experimental validation, demonstrating the method's effectiveness in improving geometric consistency across different poses.
Additional experiments on 2D images and a toy dataset further substantiate the robustness of the algorithm.

* Broader Applicability:
The potential applications of the pose classifier beyond the primary task highlight the versatility and potential impact of the proposed approach.

**Weaknesses:**

* Complexity and Scalability:
 The introduction of an auxiliary function and classifier adds complexity to the method. Details on computational efficiency and scalability in large-scale applications are lacking.

* Generalization:
While the method is effective for the specific problem addressed, there is limited discussion on its generalizability to other types of biases or different domains within 3D generation.

* Quantitative Metrics:
The paper would benefit from a more detailed presentation of quantitative metrics used to evaluate geometric consistency and rendering quality, alongside comparative analysis with existing methods.

* Background section is too long. This paper put many equations in Sec.2, but it actually did not provide strong support to the proposed uniform score distillation. Some space could be saved to explain Sec. 3.3.

**Questions:**

* How does RecDreamer handle variations in complex textures or intricate details that might not be directly related to pose?
* Can the proposed method be extended or adapted to address other biases in text-to-3D generation beyond pose inconsistency?
* What are the computational requirements for implementing RecDreamer, and how does it perform in terms of efficiency compared to baseline methods?

**Details Of Ethics Concerns:**

N.A.

---

> ### Author Response · Authors · 2024-11-24
>
> We thank Reviewer X3Gz for their valuable comments. To provide comprehensive details and visual evidence of the improvements, we have included the updated manuscript in the supplementary materials. We kindly invite you to review this version, where all new comparisons and analyses are thoroughly documented.
>
> ## Q1: "How does RecDreamer handle variations in complex textures or intricate details that might not be directly related to pose?"
> **Response:** We appreciate the reviewer’s thoughtful question about handling variations in complex textures or intricate details. RecDreamer addresses this challenge through its architecture, designed to ensure pose estimation is invariant to surface textures and fine details unrelated to pose.
>
> To achieve this invariance, we utilize a pre-trained DINOv2 [1] feature extractor, which is highly effective at separating high-level semantic content from low-level visual attributes like texture and color. In our framework, orientation score computation is performed in DINOv2’s feature space rather than the raw pixel space. This semantic-level processing inherently filters out irrelevant texture variations and intricate details, enabling the model to focus exclusively on pose-relevant structural information.
>
> Empirical results presented in **Appendix D** validate this texture invariance, demonstrating that RecDreamer consistently achieves robust pose estimation across objects with diverse surface complexities and levels of detail.
>
> - [1] Oquab et al., Dinov2: Learning robust visual features without supervision
>
> ## Q2(W2): "Can the proposed method be extended or adapted to address other biases in text-to-3D generation beyond pose inconsistency?"
> <em>[Note: Our response refers to new content in the revised manuscript:
> - Discussions: Appendix F (Page 28, Line 1560)
>     - Evidence: Figure 15 (Page 30, Line 1604)]</em>
>
> **Response:** Thank you for raising this insightful question about the broader applicability of our method. Our framework is indeed generalizable and can address various biases, provided a differentiable classifier is available to identify the bias in question.
>
> We illustrate this extensibility with an example in **Appendix F**, where we address gender bias in facial generation. Specifically, we observed that the prompt *"A person's face"* tends to produce outputs skewed towards feminine and youthful features, as shown in **Fig. 15(a)**. To mitigate this, we applied a 2D uniform score distillation algorithm using a CLIP classifier [1] with the text labels *"man"* and *"woman"* to represent the categories. The results (**Fig. 15(b)**) demonstrate that this intervention successfully diversifies the output distribution, generating faces with a broader range of gender presentations, including distinct masculine features.
>
> This experiment validates the extensibility of our method and highlights its potential for addressing other forms of bias. The key requirement is the availability of a suitable differentiable classifier for the specific bias dimension. This framework could be adapted to mitigate biases related to age, ethnicity, or stylistic preferences, offering a promising direction for future research.
>
> - [1] Radford et al., Learning transferable visual models from natural language supervision

---

> ### Author Response · Authors · 2024-11-24
>
> ## Q3(W1): "What are the computational requirements for implementing RecDreamer, and how does it perform in terms of efficiency compared to baseline methods?"
> <em>[Note: Our response refers to new content in the revised manuscript:
> - Runtime Performance: Appendix C.6 (Page 28, Line 1486)
>     - Evidence: Table 3 (Page 28, Line 1458)
> - Discussions: Appendix F (Page 28, Line 1560)]</em>
>
> **Response:** Our method currently runs on a single NVIDIA RTX 4090 GPU at a resolution of $256 \times 256$. As detailed in **Appendix C.6**, RecDreamer requires more training time compared to baselines, primarily due to the gradient back-propagation through the UNet introduced by the rectifier function.
>
> While RecDreamer has increased computational demands, it achieves significantly better geometric consistency than prior methods—a critical advancement in addressing persistent challenges in 3D generation. This quality-runtime trade-off is particularly valuable for applications where geometric accuracy is paramount.
>
> Our current implementation prioritizes theoretical validation over speed optimization, but we see clear opportunities for improvement. One promising direction is a hybrid training strategy: applying USD optimization during early epochs to establish geometry, followed by lighter computation for detail refinement. Additionally, as outlined in **Appendix F**, we analyze the computational intensity of our method and propose solutions, such as reducing redundant gradient computations through the UNet, to streamline the distillation process.
>
> In summary, we view the current computational overhead as an engineering challenge rather than a fundamental limitation. Ongoing work is focused on developing faster implementations while preserving the high quality that distinguishes RecDreamer.
>
> **Table 3:** Runtime comparison. All measurements are reported in minutes.
> | **USD** | **VSD** | **SDS** | **Debiased-SDS** | **PerpNeg** | **ESD** | **SDS-Bridge** |
> |:-------:|:-------:|:-------:|:----------------:|:-----------:|:-------:|:--------------:|
> | 297.34  | 180.70  | 43.35   | 43.58            | 45.97       | 201.55  | 73.24          |
>
> - [1] https://github.com/threestudio-project/threestudio
> - [2] McAllister et al., Rethinking Score Distillation as a Bridge Between Image Distributions
> - [3] Wang et al., Taming Mode Collapse in Score Distillation for Text-to-3D Generation

---

> ### Author Response · Authors · 2024-11-24
>
> ## W3: " ... a more detailed presentation of quantitative metrics."
> <em>[Note: Our response refers to new content in the revised manuscript:
> - Detailed Metrics: Appendix C.1 (Page 22, Line 1142)]</em>
>
> **Response:** We have restructured our discussion of evaluation metrics to improve clarity while maintaining comprehensive technical detail. The metrics are now organized into two sections: a concise overview in the main text focusing on the key roles of different losses, with detailed implementations provided in **Appendix C.1**. This revised presentation addresses several key aspects:
>
> 1. **Generation Quality (FID and uFID):**
>    We measure generation quality using two variants of the Fréchet Inception Distance (FID). The standard FID quantifies the distribution gap between rendered images and outputs from the diffusion model, indicating how closely our learned model aligns with the prior distribution's quality. However, since diffusion-generated images can exhibit inherent bias, we introduce an unbiased FID (uFID) for further evaluation. Specifically, we manually label the poses of generated images and resample them according to these pose labels, removing pose bias from the evaluation. The uFID is then calculated between the rendered images and this unbiased set.
>
> 2. **Geometric Consistency (Entropy-Based Metrics):**
>    To address geometric consistency issues, such as the Multi-Face Janus problem, we employ entropy-based metrics. These metrics operate on the principle that geometrically inconsistent models, which exhibit similar classifications across different viewpoints, will have low entropy in their averaged classification results. A lower entropy value thus indicates poorer geometric consistency, providing a targeted measure for such inconsistencies.
>
> 3. **Text-Scene Alignment (CLIP Score):**
>    Following Reviewer wW4M's suggestion, we incorporated the CLIP score [1] as an additional metric to evaluate the alignment between text prompts and the generated scene.
>
> This revised structure balances technical rigor with improved accessibility, offering a clear motivation and detailed implementation for each metric used in our evaluation.
>
> - [1] Radford et al., Learning transferable visual models from natural language supervision
>
> ## W4: " ... Background section is too long."
> **Response:** Thank you for the constructive feedback. We will streamline the Background section by concentrating on essential concepts directly relevant to our technical contributions, such as the foundations of score distillation and pose-aware generation. To improve the paper’s focus and readability, we will remove peripheral background material in the next revision.

---

### Official Review · Reviewer_1nXi · 2024-11-03

**Soundness:** 3
**Presentation:** 3
**Contribution:** 3
**Rating:** 6
**Confidence:** 3

**Summary:**

This paper proposes to rectify the biased pose distribution, ensuring the pose variation is uniformly distributed, to solve the Janus problem. A uniform score distillation module is introduced accordingly.

**Strengths:**

1. This paper rectify the biased distribution to uniform distribution by reweighing the density of original distribution.
2. A corresponding uniform score distillation process is designed to improve the consistency.

**Weaknesses:**

1. More comparison with baselines is needed, such as [1].
2. Runtime analysis is missing, the sampling time per image is expected.


[1] McAllister, David, et al. "Rethinking Score Distillation as a Bridge Between Image Distributions." arXiv preprint arXiv:2406.09417 (2024).

**Questions:**

Can the authors report the runtime per sample of your method and other baselines to ensure the completeness of comparison?

---

> ### Author Response · Authors · 2024-11-24
>
> We thank Reviewer 1nXi for their valuable comments. To provide comprehensive details and visual evidence of the improvements, we have included the updated manuscript in the supplementary materials. We kindly invite you to review this version, where all new comparisons and analyses are thoroughly documented.
>
>
> ## Q1(W2): "Can the authors report the runtime per sample of your method and other baselines to ensure the completeness of comparison?"
> <em>[Note: Our response refers to new content in the revised manuscript:
> - Runtime Performance: Appendix C.6 (Page 28, Line 1486)
>     - Evidence: Table 3 (Page 28, Line 1458)
> - Discussions: Appendix F (Page 28, Line 1560)]</em>
>
> **Response:** Detailed runtime comparisons across different methods are provided in **Appendix C.6**. Our method currently requires longer computational time due to the additional gradient back-propagation through the UNet introduced by our rectifier function. We address this runtime consideration from three key perspectives:
>
> 1. **Quality-Runtime Trade-off:**
>    While our method has a longer runtime, it achieves superior geometric consistency that is challenging to replicate with existing approaches. This trade-off is particularly valuable for applications where geometric accuracy is critical.
>
> 2. **Focus on Theoretical Validation Over Optimization:**
>    Our current implementation prioritizes theoretical validation rather than computational efficiency. However, there are several promising directions for runtime improvement:
>    - Implementing a hybrid optimization strategy: employing USD during the initial epochs to establish geometric consistency, followed by lightweight optimization for detail refinement.
>    - Optimizing implementation: Our method currently lacks some of the optimization tricks used in other three-studio-based approaches [1], such as SDS-Bridge [2] and ESD [3], which employ techniques like low-resolution warmup. Incorporating these practices could significantly reduce runtime without sacrificing performance.
>
> 3. **Future Algorithmic Improvements:**
>    As outlined in **Appendix F**, potential enhancements to our method include optimizing the conditional gradient back-propagation through the UNet to minimize computational overhead while maintaining high-quality outputs.
>
> These considerations highlight both the strengths of our method and the clear pathways for reducing runtime in future work.
>
> **Table 3:** Runtime comparison. All measurements are reported in minutes.
> | **USD** | **VSD** | **SDS** | **Debiased-SDS** | **PerpNeg** | **ESD** | **SDS-Bridge** |
> |:-------:|:-------:|:-------:|:----------------:|:-----------:|:-------:|:--------------:|
> | 297.34  | 180.70  | 43.35   | 43.58            | 45.97       | 201.55  | 73.24          |
>
> - [1] https://github.com/threestudio-project/threestudio
> - [2] McAllister et al., Rethinking Score Distillation as a Bridge Between Image Distributions
> - [3] Wang et al., Taming Mode Collapse in Score Distillation for Text-to-3D Generation

---

> ### Author Response · Authors · 2024-11-24
>
> ## W1: "More comparison with baselines is needed."
> <em>[Note: Our response refers to new content in the revised manuscript:
> - Main experiment: Table 1 (Page 8, Line 378)
> - User study: Table 4 (Page 28, Line 1463)
> - Results: Figure 17 (Page 32, Line 1727)]</em>
>
> **Response:** We sincerely thank the reviewer for this constructive suggestion, which has significantly improved our evaluation framework. In response, we have expanded our experimental comparisons to include additional state-of-the-art baselines, specifically SDS-Bridge [1] and ESD [2]. The comprehensive results of these comparisons are presented in **Table 1** and **Table 4**, with visual examples provided in **Fig. 17**. These additions enhance the robustness of our evaluation and offer a more complete context for assessing our method's contributions relative to existing approaches.
>
> **Table 1:** Quantitative comparison.
> | **Method** | **FID (↓)** | **uFID (↓)** | **cEnt (↑)** | **pEnt (↑)** | **CLIP (↓)** |
> |:------------:|:-----------:|:------------:|:------------:|:------------:|:------------:|
> | SDS          | 204.81      | 205.66       | 1.0235       | 1.1542       | 0.6966       |
> | Debiased-SDS | 219.46      | 218.83       | 1.0171       | 1.0609       | 0.7251       |
> | PerpNeg      | 203.01      | 203.45       | 1.0348       | 1.0390       | 0.7076       |
> | ESD          | 187.31      | 188.13       | 1.0271       | 1.0928       | 0.6871       |
> | SDS-Bridge   | 230.87      | 229.41       | 1.0278       | 1.0932       | 0.7250       |
> | VSD          | 168.19      | 169.66       | 1.0276       | 1.0676       | 0.6807       |
> | USD          | 165.97      | 165.25       | 1.0375       | 1.2488       | 0.6842       |
>
>
> **Table 4:** User study on geometric consistency. Higher indicates better.
> | **USD** | **VSD** | **SDS** | **Debiased-SDS** | **PerpNeg** | **ESD** | **SDS-Bridge** |
> |:-------:|:-------:|:-------:|:----------------:|:-----------:|:-------:|:--------------:|
> | 4.80    | 3.27    | 2.81    | 2.44             | 1.96        | 3.21    | 2.50           |
>
> - [1] McAllister et al., Rethinking Score Distillation as a Bridge Between Image Distributions
> - [2] Wang et al., Taming Mode Collapse in Score Distillation for Text-to-3D Generation

---

> ### Author Response · Authors · 2024-11-24
>
> We have carefully addressed the main concerns raised in the review.
>
> First, we conducted a thorough evaluation of runtime performance. While our method incurs an increase in runtime, it effectively mitigates the Multi-Face Janus problem, representing a valuable trade-off in applications where geometric consistency is critical. Additionally, we have proposed potential solutions to reduce runtime in future iterations.
>
> Second, we have expanded our comparisons to include the suggested SDS-Bridge method, incorporating relevant discussions into sections such as related work.
>
> Given these efforts, we respectfully hope the reviewer might reconsider their assessment of our paper’s contributions. We believe these improvements have significantly strengthened our work and addressed the key concerns outlined in the initial review.

---

> > ### Comment · Reviewer_1nXi · 2024-11-26
> >
> > Thank you for your detailed response to my previous comments. After carefully reviewing your rebuttal, I believe my concerns have been addressed, and I have raised my score accordingly.
> >
> > Strengths:
> > 1. The authors have conducted comprehensive experiments to validate the effectiveness of the proposed USD.
> > 2. The generation quality surpasses other baselines, even though the speed is not state-of-the-art—a common limitation for optimization-based methods.
> >
> > Recommendations:
> > 1. Consider including a runtime comparison in the manuscript for greater clarity.
> > 2. Ensure the timely release of the code to enhance reproducibility.
> >
> > Thank you once again for your efforts. I believe this paper meets the standards for publication at ICLR.

---

> > > ### Author Response · Authors · 2024-11-27
> > >
> > > Thanks for the detailed review and positive feedback! We'll revise the paper to include the runtime comparison in the Appendix and release the code soon to support reproducibility.

---

### Official Review · Reviewer_wW4M · 2024-11-04

**Soundness:** 4
**Presentation:** 4
**Contribution:** 3
**Rating:** 8
**Confidence:** 5

**Summary:**

This paper tackles the Janus problem with score distillation-based text-to-3D generation via debiasing the camera pose distribution of the pre-trained text-to-image diffusion model. Specifically, the authors derive an auxiliary function that rectifies the pose distribution induced by the original image distribution to a targeted distribution (e.g. uniform). Consequently, the score distillation rule yielded from the debiased image distribution, termed uniform score distillation (USD), is derived as a combination of the variational score distillation plus the “score” of the auxiliary function. The auxiliary rectifier can be computed with a pose classifier and a running estimator of the pose distribution over the Gaussian perturbed image distribution.

**Strengths:**

+ This paper offers a distinctive approach to addressing the Janus problem in score distillation-based text-to-3D generation. Its central argument asserts that the pre-trained image distribution is intrinsically biased toward canonical views, causing the resulting 3D content to exhibit this same bias—manifesting as the Janus problem in practical applications. This perspective contrasts with most existing score distillation methods, which often assume that the 2D pre-trained distribution is ideal and strive to align the view distribution with it more closely.
+ The proposed method is both novel and theoretically grounded, offering a unified and principled approach to debiasing that integrates seamlessly into current score distillation frameworks. The debiasing mechanism is based on solid theoretical underpinnings yet is surprisingly straightforward to implement. All derivations appear correct. The proposed estimation for the otherwise intractable auxiliary rectifier term is thoughtfully designed, with each step justified through theoretical insight.
+ The paper is dense and well-written, with thorough engineering details, including but not limited to an efficient pose classifier, a running estimator of the pose distribution, and improved training techniques. The results are extensive: beyond fundamental qualitative outcomes, the paper includes numerous validation experiments in the appendix, demonstrating the efficacy of each design element.

**Weaknesses:**

- The quantitative results in this paper are somewhat limited. While I recognize the lack of a standard metric for evaluating generated 3D results, a few commonly used measures, such as the CLIP score demonstrated in SDS [1], could still be applied. Given the high degree of randomness often seen in score-distilled outputs, it would also be beneficial to assess the method using human evaluation, specifically by measuring the success rate of Janus-free results [2].

- Another limitation is that the proposed USD appears to require reference images from various view angles, which may be impractical when the prompt is abstract or imaginary and lacks similar images on the shelf. The authors may wish to discuss potential solutions to this constraint.

- The paper also overlooks several related works in the text-to-3D generation literature. While the field is vast, I suggest including a more comprehensive review. Some key omissions in the current version include: [3,4,5,6]

- Many state-of-the-art approaches currently rely on training or fine-tuning generative models on 3D data. It would be interesting to show whether score distillation with the rectified distribution could largely match the performance of the distribution fine-tuned with extra data. I’d also suggest demonstrate whether the proposed USD can even enhance these training-based methods.


[1] Poole et al., DreamFusion: Text-to-3D using 2D Diffusion

[2] Wang et al., Taming Mode Collapse in Score Distillation for Text-to-3D Generation

[3] Yu et al., Text-to-3D with Classifier Score Distillation

[4] Katzir et al., Noise-free Score Distillation

[5] Wang et al., SteinDreamer: Variance Reduction for Text-to-3D Score Distillation via Stein Identity

[6] Shi et al., MVDream: Multi-view Diffusion for 3D Generation

**Questions:**

1. Why is the VSD loss in Eq. 14 denoted as $L’_{VSD}$ instead of $L_{VSD}$​ as in Eq. 9?

2. Additionally, I wonder whether modifying the camera sampling distribution in VSD to use the estimated $p(c∣y)$ could yield similar or complementary effects in mitigating Janus problems.

---

> ### Author Response · Authors · 2024-11-24
>
> We thank Reviewer wW4M for their valuable comments. To provide comprehensive details and visual evidence of the improvements, we have included the updated manuscript in the supplementary materials. We kindly invite you to review this version, where all new comparisons and analyses are thoroughly documented.
>
> ## Q1: "Why is the VSD loss in Eq. 14 denoted as $L'\_{VSD}$ instead of $L_{VSD}$ as in Eq. 9?"
> **Response:** The distinction between $L'\_{VSD}$ and $L\_{VSD}$ highlights a key difference in the treatment of directional prompts. In the original $L\_{VSD}$ formulation (Eq. 9), the text conditioning adapts to the camera pose, utilizing specific directional prompts (e.g., "front view," "side view") for each sampling position $q$. In contrast, the modified $L'\_{VSD}$ (Eq. 14) employs a single, fixed prompt that remains consistent across all camera poses. This simplification is denoted by omitting the directional conditioning subscript $c$ from the loss term in the modified formulation.
>
> ## Q2: " ... modifying the camera sampling distribution in VSD to use the estimated p(c∣y) could yield similar or complementary effects?"
> <em>[Note: Our response refers to new content in the revised manuscript:
> - Related works: Appendix C.2 (Page 23, Line 1235)
> - Qualitative results: Figure 9 (Page 25, Line 1306)]</em>
>
> **Response:** We thank the reviewer for the insightful suggestion regarding the modification of the camera sampling distribution in VSD using the estimated $p(c|y)$. While we explored this direction extensively, as detailed in **Appendix C.2** (Sampling $q$), our experiments indicate that such modifications do not result in the anticipated improvements in 3D consistency. Our analysis highlights two fundamental limitations:
>
> 1. **Data Bias in the Underlying Prior:**
>    Adjusting the view sampling probabilities in the $q$ distribution does not address the inherent data bias in the training prior. Certain viewpoints are naturally underrepresented in the training data, and this scarcity remains a limiting factor even with modified sampling strategies.
>
> 2. **Optimization Challenges from Focused Sampling:**
>    Experiments show that concentrating sampling on specific viewpoints introduces problematic optimization behavior. Specifically, we observe that scenes become flatter when optimization focuses too heavily on certain camera poses, leading to difficulties in convergence. Furthermore, the lack of comprehensive viewpoint supervision prevents the model from synthesizing geometrically coherent structures.
>
> These findings suggest that the challenges inherent in the training prior and optimization dynamics cannot be resolved solely through modifications to the $q$ distribution sampling strategy. The quantitative results supporting this analysis are presented in **Fig. 9**.

---

> ### Author Response · Authors · 2024-11-24
>
> ## W1: "CLIP score and human evaluation of success rate."
> <em>[Note: Our response refers to new content in the revised manuscript:
> - Main experiment: Table 1 (Page 8, Line 378)
> - Success Rate: Appendix C.7 (Page 28, Line 1499)
>     - Evidence: Table 5 (Page 28, Line 1470)
> - Discussions: Appendix F (Page 28, Line 1560)]</em>
>
> **Response:** Thank you for your valuable feedback regarding evaluation metrics. We have addressed both points as follows:
>
> 1. **CLIP Scores [1] (Shown in **Table 1**):**
>    We have included CLIP score evaluations in the main paper. While CLIP scores measure image-text consistency, it is important to note that our method generates multi-view results, including viewpoints (e.g., back views) that may not be explicitly described in the prompt. This viewpoint diversity enhances 3D consistency but may lead to lower CLIP scores since the additional views do not strictly align with the text description.
>
> 2. **Success Rate Evaluation (Shown in **Table 5**):**
>    In response to your insightful suggestion regarding human evaluation, we conducted a detailed manual assessment (described in **Appendix C.7**) using a structured scoring system. Specifically, we assigned a score of 1.0 for global feature inconsistency and 0.5 for local feature inconsistencies (e.g., in features like ears, hands, and feet). The results show that our method effectively resolves global inconsistencies, although challenges persist in maintaining fine-grained detail consistency. To address these challenges, future work could explore incorporating multi-view image reconstruction supervision techniques to better constrain the score distillation process. Further details on this potential direction are discussed in **Appendix F**.
>
> **Table 1:** Quantitative comparison.
> | **Method** | **FID (↓)** | **uFID (↓)** | **cEnt (↑)** | **pEnt (↑)** | **CLIP (↓)** |
> |:------------:|:-----------:|:------------:|:------------:|:------------:|:------------:|
> | SDS          | 204.81      | 205.66       | 1.0235       | 1.1542       | 0.6966       |
> | Debiased-SDS | 219.46      | 218.83       | 1.0171       | 1.0609       | 0.7251       |
> | PerpNeg      | 203.01      | 203.45       | 1.0348       | 1.0390       | 0.7076       |
> | ESD          | 187.31      | 188.13       | 1.0271       | 1.0928       | 0.6871       |
> | SDS-Bridge   | 230.87      | 229.41       | 1.0278       | 1.0932       | 0.7250       |
> | VSD          | 168.19      | 169.66       | 1.0276       | 1.0676       | 0.6807       |
> | USD          | 165.97      | 165.25       | 1.0375       | 1.2488       | 0.6842       |
>
> **Table 5:** Success rates for Janus-free generation
> | **Prompts** | **Mean** | **Std** | **Median** | **Mode** | **Min** | **Max** |
> |:-----------:|:--------:|:-------:|:----------:|:--------:|:-------:|:-------:|
> | kangaroo    | 0.65     | 0.5296  | 0.5        | 0.5      | 0       | 1.5     |
> | bear        | 0.94     | 0.5270  | 0.75       | 0.5/1    | 0.5     | 2       |
>
> - [1] Radford et al., Learning transferable visual models from natural language supervision

---

> ### Author Response · Authors · 2024-11-24
>
> ## W2: " ... potential solutions to abstract or imaginary prompts."
> <em>[Note: Our response refers to new content in the revised manuscript:
> - Template images: Appendix C.3 (Page 26, Line 1387)
>     - Supporting evidence: Figure 10 (Page 25, Line 1333)
> - Cross-modality: Appendix C.4 (Page 26, Line 1397)]</em>
>
> **Response:** Thank you for raising this important question regarding the handling of abstract and imaginary prompts. Our research explores innovative strategies to address such prompts, with detailed findings provided in **Appendix C.3**. The key components of our approach are as follows:
>
> 1. **Generating Abstract and Imaginative Templates:**
>    As demonstrated in **Fig. 10**, we generate reference template images using Stable Diffusion [1]. These generated images effectively serve as templates, allowing users to select representative examples for different poses. We have also implemented an automated pipeline where the diffusion model first generates concept images in a canonical pose, which are subsequently processed by our multi-view model [2] to synthesize coherent perspectives from diverse angles.
>
> 2. **Adaptability of Template Requirements:**
>    As shown in **Fig. 10**, the template images used in our pipeline do not need to be multi-view consistent or high-quality. They only require basic orientation information, making the approach highly adaptable across various applications. This flexibility allows for practical deployment in handling abstract or low-quality inputs.
>
> 3. **Cross-Modal and Cross-Category Capabilities:**
>    Our approach incorporates the powerful feature matching capabilities of DINOv2 [3], enabling robust cross-modal generation. This allows the system to work with diverse inputs, such as sketches and stylistically varied images, and even perform cross-category generation (e.g., using cat images to generate dogs). The generated outputs maintain semantic consistency while allowing for creative transformations. An application of sketch-based generation is detailed in **Appendix C.4**.
>
> These features, combined with examples extensively documented in **Appendix C.3**, showcase the ability of our method to handle abstract and imaginative prompts while ensuring consistent multi-view generation.
>
> - [1] Rombach et al., High-resolution image synthesis with latent diffusion models
> - [2] Liu et al., Zero-1-to-3: Zero-shot one image to 3d object
> - [3] Oquab et al., Dinov2: Learning robust visual features without supervision
>
> ## W3: " ... a more comprehensive review."
> <em>[Note: Our response refers to new content in the revised manuscript:
> - Related works: Appendix A (Page 15, Line 756)]</em>
>
> **Response:** We appreciate the reviewer's suggestion regarding the literature review. We have substantially expanded our discussion of related works in **Appendix A**, incorporating several recent and significant contributions to the field. The additions include pivotal works on score distillation techniques as follows:
> - [1] Yu et al., Text-to-3D with Classifier Score Distillation
> - [2] Katzir et al., Noise-free Score Distillation
> - [3] Wang et al., SteinDreamer: Variance Reduction for Text-to-3D Score Distillation via Stein Identity
> - [4] Shi et al., MVDream: Multi-view Diffusion for 3D Generation
> - [5] McAllister et al., Rethinking Score Distillation as a Bridge Between Image Distributions
> - [6] Jiang et al., JointDreamer: Ensuring Geometry Consistency and Text Congruence in Text-to-3D Generation via Joint Score Distillation

---

> ### Author Response · Authors · 2024-11-24
>
> ## W4-1: " ... whether score distillation with the rectified distribution could largely match the performance of the distribution fine-tuned with extra data."
> <em>[Note: Our response refers to new content in the revised manuscript:
> - 3D Generation: Appendix F (Page 29, Line 1560)
>     - Evidence: Figure 16 (Page 30, Line 1632)]</em>
>
> **Response:** We thank the reviewer for raising this important question. Our comparative analysis highlights a trade-off between geometric consistency and stylistic flexibility in methods based on score distillation, including ours, compared to approaches trained on 3D data.
>
> To illustrate this trade-off, we present an experiment in **Appendix F** using a challenging prompt: *"A platypus, dressed in a video game pixelated costume, steps on a pixelated surfboard and holds a squid weapon that emits 8-bit light effects."* As shown in **Fig. 16(a)**, 3D-trained methods like Tripo AI [1] excel at producing precise geometry but struggle to capture stylistic elements, such as the "pixelated" and "8-bit" effects described in the prompt. In contrast, our approach (**Fig. 16(b)**) successfully captures these stylistic nuances, including the digital effects, but demonstrates lower geometric consistency.
>
> This comparison underscores the complementary strengths of these methods. While 3D-trained approaches prioritize geometric precision, score distillation methods offer greater flexibility in generating stylized outputs. These findings suggest promising directions for future work that could integrate the advantages of both approaches to achieve robust geometric consistency and stylistic versatility.
>
> - [1] https://lumalabs.ai/genie
>
> ## W4-2: " ... whether the proposed USD can enhance training-based methods."
> <em>[Note: Our response refers to new content in the revised manuscript:
> - 3D Generation: Appendix F (Page 29, Line 1560)
>     - Evidence: Figure 16 (Page 30, Line 1632)]</em>
>
> **Response:** Thank you for this insightful comment on USD's potential to enhance training-based methods. Our experimental results in **Appendix F** demonstrate that USD can effectively leverage the 2D prior to improve visual quality. Specifically, as shown in **Fig. 16**, applying USD with templates generated by Tripo AI [1] leads to meaningful enhancements in the output's visual fidelity.
>
> While training from scratch may introduce some limitations in geometric consistency, these challenges can be addressed in future work through direct fine-tuning on 3D models, providing a promising direction for further refinement.
>
> - [1] https://lumalabs.ai/genie

---

> ### Comment · Reviewer_wW4M · 2024-12-03
>
> Thanks authors for the extensive clarifications. Most of my concerns are addressed. I keep my score and remain positive for this work.

---

> > ### Author Response · Authors · 2024-12-04
> >
> > We appreciate reviewer wW4M's acknowledgment of our contributions.

---

### Official Review · Reviewer_i678 · 2024-11-04

**Soundness:** 2
**Presentation:** 3
**Contribution:** 3
**Rating:** 6
**Confidence:** 4

**Summary:**

The authors of the paper introduce RecDreamer, a text-to-3D generation task that aims to reshape the underlying data distribution of pretrained text-to-image diffusion models to eliminate the Multi-face Janus problem. To achieve this, they develop an auxiliary function derived from the joint distribution expression of the data x and camera pose c. A key component of this auxiliary function is a well-designed, lightweight pose classifier capable of calculating p_t(c|x_0). Extensive experiments demonstrate the effectiveness of RecDreamer in eliminating the Janus problem.

**Strengths:**

1)	The authors present an insightful analysis of the emergence of the multi-face Janus problem in text-to-3D generation. Building on their analysis, they offer a solid theoretical framework on reshaping the underlying data distribution of pretrained text-to-image diffusion models.
2)	The design of the pose classifier appears to be innovative, with clear and comprehensive details provided in the appendix.
3)	The qualitative and quantitative results of RecDreamer demonstrate its state-of-the-art performance, as validated in their experiment section.

**Weaknesses:**

1)	The evaluation of the paper is limited. Initially, all experiments are only tested on 22 prompts from the original DreamFusion gallery, while previous works typically use over 40. The prompt list is also not provided in the appendix, which compromises the reproducibility of the experiments. Furthermore, the absence of recent open-sourced baseline methods that focus on addressing the Janus problem, such as ESD1 and JointDreamer2, is notable.
2)	The authors introduce the concept of the "joint distribution expression of the data and camera pose," but provide limited explanation of the practical meaning of such a distribution. Also, the use of a rather discrete 4-views to represent the camera pose distribution does not appear to be meaningful. More ablation studies should be conducted.
3)	The authors state that the image templates are "user-provided" in line 317 of the main manuscript, which seems unusual. It is not feasible for users to provide consistent multi-view templates for any text prompts.

**Questions:**

I would appreciate it if the authors could provide a clear explanation of the second and third weaknesses. I am open to revising my score if the author addresses this concern.

---

> ### Author Response · Authors · 2024-11-24
>
> We thank Reviewer i678 for their valuable comments. To provide comprehensive details and visual evidence of the improvements, we have included the updated manuscript in the supplementary materials. We kindly invite you to review this version, where all new comparisons and analyses are thoroughly documented.
>
> ## Q1(W2)-1: " ... joint distribution expression of the data and camera pose, limited explanation of the practical meaning of such a distribution."
> **Response:** We thank the reviewer for raising this important point about the interpretation of the joint distribution. We provide the following detailed explanation to clarify its practical meaning:
>
> The joint distribution $p(x, c)$ represents the probability of observing both a specific data point $x$ and a camera pose $c$ simultaneously. This formulation serves as a crucial mathematical connection between the data distribution $p(x)$ and the pose distribution $p(c)$. Without this joint formulation, these two distributions would remain independent, making it challenging to analyze or model their interdependence. The joint distribution enables us to derive the marginal distributions by integration: $p(c)$ can be obtained by marginalizing over $x$ ($\int p(x, c) dx$), and $p(x)$ can be obtained by marginalizing over $c$ ($\int p(x, c) dc$). This approach is particularly valuable because it allows us to impose specific constraints on the marginal distributions, such as enforcing $p(c)$ to follow a uniform distribution, while preserving the mathematical relationship between data and pose.
>
> ## Q1(W2)-2: "Also, the use of a rather discrete 4-views to represent the camera pose distribution does not appear to be meaningful."
> **Response:** We thank the reviewer for their thoughtful observation regarding our representation of the camera pose distribution. The use of four discrete perspectives in our design effectively simulates the global pose distribution, and this choice is supported by two key considerations:
>
> 1. **Robustness of Score Distillation to Coarse Pose Supervision:**
>    Score distillation with diffusion models is inherently robust to coarse pose supervision. Our four-view approach (front-back-left-right) provides sufficient semantic guidance for the task. Unlike geometric reconstruction methods [1,2], which require precise pose estimation, score distillation relies primarily on general semantic cues to transfer pre-trained knowledge into 3D models. This robustness is evidenced by prior works such as Vanilla SDS [3] and VSD [4], which achieve effective optimization using similarly broad pose categories (e.g., "front view," "back view," "side view").
>
> 2. **Objective to Mitigate Pose Bias, Not Precise Pose Control:**
>    Our primary goal is to address pose bias by enabling the generation of diverse poses, rather than achieving precise pose control. The four-view discretization effectively captures sufficient pose variations guided by the templates during score distillation. This approach strikes a balance between simplicity of implementation and the robustness needed for our objectives.
>
> These considerations demonstrate that the four-view discretization represents a practical and effective design choice, ensuring diversity in pose expression while maintaining robust performance.
>
> - [1] Mildenhall et al., NeRF: Representing Scenes as Neural Radiance Fields for View Synthesis
> - [2] Kerbl et al., 3D Gaussian Splatting for Real-Time Radiance Field Rendering
> - [3] Poole et al., DreamFusion: Text-to-3D Using 2D Diffusion
> - [4] Wang et al., ProlificDreamer: High-Fidelity and Diverse Text-to-3D Generation with Variational Score Distillation

---

> ### Author Response · Authors · 2024-11-24
>
> ## Q1(W3): " ... not feasible for users to provide consistent multi-view templates for any text prompts."
> <em>[Note: Our response refers to new content in the revised manuscript:
> - Template images: Appendix C.3 (Page 26, Line 1387)
>     - Supporting evidence: Figure 10 (Page 25, Line 1333)
> - Cross-modality: Appendix C.4 (Page 26, Line 1397)]</em>
>
> **Response:** We appreciate the reviewer’s concerns and would like to emphasize that our method imposes minimal requirements on template images, both in terms of quantity and quality. Specifically:
>
> 1. **Flexibility in Template Requirements:**
>    Our approach does not require the template images to be view-consistent or even from the same modality, making it highly practical for real-world applications. As shown in **Appendix C.3** (**Fig. 10(a)**), we use Stable Diffusion [1] outputs as templates, without demanding high-quality or view-consistent images.
>
> 2. **Support for Cross-Modality Inputs:**
>    We further demonstrate the flexibility of our method by supporting multiple input modalities. For instance, we achieve successful results using hand-drawn sketches for cross-modality generation, as detailed in **Appendix C.4**. Moreover, while not necessary, users may optionally employ multi-view image generation models [2] to generate additional viewpoints if desired, as illustrated in **Fig. 10(b)**.
>
> These features make our classifier-based approach lightweight and practical, requiring minimal effort from users while maintaining robust and versatile performance.
>
> - [1] Rombach et al., High-resolution Image Synthesis with Latent Diffusion Models
> - [2] Liu et al., Zero-1-to-3: Zero-shot One Image to 3D Object
>
> ## W1-1: " ... prompt list is also not provided in the appendix ... "
> <em>[Note: Our response refers to new content in the revised manuscript:
> - Prompt list: Appendix G (Page 31, Line 1653)]</em>
>
> **Response:** Thank you for the suggestion. We have provided in the **Appendix G** in the revised version.
>
> ## W1-2: " ... recent open-sourced baseline methods ESD and JointDreamer."
> <em>[Note: Our response refers to new content in the revised manuscript:
> - Related works: Appendix A (Page 15, Line 756)
> - Main experiment: Table 1 (Page 8, Line 378)
> - User study: Table 4 (Page 28, Line 1463)]</em>
>
> **Response:** Thank you for your suggestion. We have expanded our evaluation by including additional baseline methods, specifically ESD [1] and Bridge-SDS [2], which rely solely on diffusion models without requiring 3D data for fine-tuning. The quantitative results and findings from the user study are presented in **Table 1** and **Table 4**. Regarding JointDreamer [3], while it is an important contribution in this domain, we discuss it in our related work section (**Appendix A**) due to its fundamentally different approach, which relies on view-aware pre-trained models.
>
> **Table 1:** Quantitative comparison.
> | **Method** | **FID (↓)** | **uFID (↓)** | **cEnt (↑)** | **pEnt (↑)** | **CLIP (↓)** |
> |:------------:|:-----------:|:------------:|:------------:|:------------:|:------------:|
> | SDS          | 204.81      | 205.66       | 1.0235       | 1.1542       | 0.6966       |
> | Debiased-SDS | 219.46      | 218.83       | 1.0171       | 1.0609       | 0.7251       |
> | PerpNeg      | 203.01      | 203.45       | 1.0348       | 1.0390       | 0.7076       |
> | ESD          | 187.31      | 188.13       | 1.0271       | 1.0928       | 0.6871       |
> | SDS-Bridge   | 230.87      | 229.41       | 1.0278       | 1.0932       | 0.7250       |
> | VSD          | 168.19      | 169.66       | 1.0276       | 1.0676       | 0.6807       |
> | USD          | 165.97      | 165.25       | 1.0375       | 1.2488       | 0.6842       |
>
>
> **Table 4:** User study on geometric consistency. Higher indicates better.
> | **USD** | **VSD** | **SDS** | **Debiased-SDS** | **PerpNeg** | **ESD** | **SDS-Bridge** |
> |:-------:|:-------:|:-------:|:----------------:|:-----------:|:-------:|:--------------:|
> | 4.80    | 3.27    | 2.81    | 2.44             | 1.96        | 3.21    | 2.50           |
>
>
> - [1] Wang et al., Taming Mode Collapse in Score Distillation for Text-to-3D Generation
> - [2] McAllister et al., Rethinking Score Distillation as a Bridge Between Image Distributions
> - [3] Jiang et al., JointDreamer: Ensuring Geometry Consistency and Text Congruence in Text-to-3D Generation via Joint Score Distillation

---

> ### Author Response · Authors · 2024-11-24
>
> We sincerely thank Reviewer i678 for the constructive suggestions. We believe that the additional experiments, analysis, and explanations have significantly enhanced the quality of our submission. We hope these improvements provide sufficient justification for a higher score.

---

> > ### Comment · Reviewer_i678 · 2024-11-27
> > **Thanks!**
> >
> > Thanks for the authors' efforts during the rebuttal period; however, my concerns remain unresolved. The authors mentioned that JointDreamer "relies on view-aware pre-trained models," which is why they did not provide a direct comparison with it. However, since RecDreamer also requires users to generate template images using models like Stable Diffusion or Zero-1-to-3, I am unsure why MVDream, similar to JointDreamer, was not utilized to enable a fair comparison. As this concern has not been fully addressed, I have decided to maintain my original rating.

---

> > > ### Author Response · Authors · 2024-11-27
> > >
> > > Thank you for your continued feedback and careful review of our work. We appreciate the opportunity to further clarify our approach and address your concerns.
> > > Regarding the comparison with JointDreamer and the use of multi-view diffusion models, we would like to provide additional context:
> > >
> > > 1. **Minimal View-Aware Diffusion Dependency of RecDreamer**
> > >
> > >     Compared with methods like JointDreamer, our RecDreamer does not rely on view-aware diffusions.
> > >
> > >     By default, RecDreamer relies on the *original Stable Diffusion model*. Our mention of Zero-1-to-3 and sketch templates is solely to demonstrate the method's flexibility, **not to establish a core dependency on multi-view diffusion models**.
> > >
> > >     While users may have the option to use Zero-1-to-3 to generate images from four views, our approach utilizes Zero-1-to-3 differently from JointDreamer, operating under distinct generative settings. JointDreamer focuses on aggregating a 3D model from inconsistent view information across different poses, whereas RecDreamer aims to achieve pose-uniform distributions without relying on pose-specific constraints. Consequently, these methods are inherently difficult to compare due to their fundamentally different objectives and setups.
> > >
> > >
> > >
> > > 2. **Challenges in Fair Comparative Evaluation**
> > >
> > >     The core challenge in comparing methods like RecDreamer and JointDreamer lies in their fundamentally different score functions. These differences significantly impact generation capabilities:
> > >     - RecDreamer relies entirely on the *original Stable Diffusion*'s prior knowledge
> > >     - Methods like JointDreamer use *view-aware diffusion* models designed for specific pose generation
> > >
> > >     This architectural difference makes direct comparisons challenging. For instance, methods optimized on 3D datasets like Objaverse may excel with object-specific prompts but potentially struggle with more imaginative or out-of-domain prompts.
> > >
> > > We acknowledge the complexity of establishing a truly fair comparative framework and remain open to constructive suggestions for more comprehensive evaluation methodologies.

---

### Public Comment · ~Bo_Yu9 · 2024-11-27
**Several question about the  calculation of orientaton distance and EMA version of the initial pose probability**

Thank you for your interesting work! It is well-written, clearly structured, and has enlightened me a lot. However, I have several questions about the implementation details in your paper and couldn’t wait to reach out before it is published.

1.Appendix B.1.2
Figure 5 is confusing to me. How are $\hat{m}_u$, {$\hat{m}^{i_2}_u$} and {$\hat{m}^{i_3}_u$} obtained? Could you please provide more details on the calculation process?

2.Appendix B.1.3
> Subsequently, based on the cross-attention feature map, we can infer the foreground-background meaning for the first component.

How does this process work? What is the pipeline for mask segmentation?

3.Appendix B.2.4
> The EMA version of the pose probability, $\hat{p}_t(\hat{c}|y)$, is then given by $\hat{p}_t(\hat{c}|y)=e^{\lfloor t / n_s \rfloor}$.

Is this an initialization value for the pose probability? Why is it not set to $1/n_p$, where $n_p$ is the number of pose categories?

If my questions are inconvenient or if you do not have the time to address them at the moment, please feel free to disregard this message.

Thank you for your time and consideration.

Best regards.

---

> ### Author Response · Authors · 2024-11-27
>
> Thanks for your attention. Since Q1 and Q3 need further clarification, we will provide more details later.
>
> ## Q2 "How does this process work? What is the pipeline for mask segmentation?"
> **Response:** This is actually a trick mentions in DINOv2 [1]. To segment the images, the reviewers construct a PCA to classify the patch tokens. Specifically, images are encoded as patch features (with tensor shape $[B, C, H\*W]$). Then, PCA is applied on all the patch features ($[B\*H\*W, C]$, performed on the feature channel $C$). The first channel in the compressed feature ($[B\*H\*W, C']$, where $C' << C$) is the most distinctive component, which indicates the foreground and background in the DINO feature space. However, PCA does not specify whether the value $>0$ or $<0$ represents foreground, so we add a simple postprocessing step as discussed in Appendix B.1.3.
>
> - [1] Oquab et al., Dinov2: Learning robust visual features without supervision

---

> > ### Public Comment · ~Bo_Yu9 · 2024-11-28
> > **Thanks for your reply!**
> >
> > Thanks for your reply! And i have another question, i hope i didn't bother you too much:
> >
> > 4: Is VSD neccessary for your framework to solve the Janus problem? Or it is just for the improvement for quality?

---

> > > ### Author Response · Authors · 2024-11-28
> > >
> > > ## Q1 "How are $\hat{m}\_u$, $\{\hat{m}\_u^{i\_2}\}$, and $\{\hat{m}\_u^{i\_3}\}$ obtained?"
> > > **Response:** The previous captions of Fig. 5(h) and 5(m) are too simplified. We have revised it and would like to provide a detailed clarification.
> > >
> > > Fig. 5 is a visualization of the orientation distance between the patch $u$ of input $x$ and all the patches of the template images $x^{i_2}$ (Fig. 5(e)(f)(g)(h)) and $x^{i_3}$ (Fig. 5(j)(k)(l)(m)).
> > >
> > > - $\hat{m}_u$ (Fig. 5(d)) represents the value specific patch $u$ of the input $x$ that we are calculating.
> > > - Fig. 5(h) and 5(m) show the similarity between $\hat{m}\_u$ and each patch in $\hat{m}^{i\_2}$/$\hat{m}^{i\_3}$ (each grid represent $s_{u, u'}$). The similarity for each patch $u'$ in $\hat{m}^{i\_2}$/$\hat{m}^{i\_3}$ is calculated by $s\_{u, u'}= 1 - \sigma\_{\tau\_{pat}}(\|\hat{\boldsymbol{f}}\_{pat, u} - \hat{\boldsymbol{f}}\_{pat, u'}^i\|\_2)$.
> > >
> > >
> > > ## Q3 "Is this an initialization value for the pose probability? Why is it not set to $1/\boldsymbol{n_p}$, where $\boldsymbol{n_p}$ is the number of pose categories?"
> > > **Response:** In the context, $e$ does not represent an exponential operation, but the elements in the list of EMA. The following is a more clarified description:
> > >
> > >
> > > We maintain a list of Exponential Moving Average (EMA) values, $\\{e^i\\}\_{i=0}^{n\_t}$, where the index is determined by dividing the original time $t$ into intervals of size $n_s$. Since the original time $t$ ranges from 0 to 1000, we group consecutive $n_s$ values together. Specifically, the EMA value for a given time $t$ is selected by $\bar{p}_t(\bar{c}|y) = e^{\lfloor t/n_s \rfloor}$, which means the same EMA value is used for all times within each interval (e.g., $e^0$ for times $[0, 100)$, $e^1$ for times $[100, 200)$, and so on).
> > >
> > > This will be updated later to avoid interruption of the current line number for the reviews.
> > >
> > >
> > > ## Q4 "Is VSD necessary for your framework to solve the Janus problem? Or it is just for the improvement for quality?"
> > > **Response:** VSD is not necessary, but the usage of VSD is not simply for improving the quality. The intuitive is that the target of VSD is to approximate the *complete prior distribution*, i.e., optimize with the objective $D_{\mathrm{KL}}(q_t^\mu(x_t|c,y)\parallel p_t(x_t|y^c))$. Under these circumstances, we further talk about "rectifying the prior distribution". However, methods like SDS are empirically not a distributional approximation, but degrades to mode seeking, which tends to approximate only the peak. This mode-seeking property might probably result in some failures in the USD algorithm as it disregards the non-peak probability.
> > >
> > > Thank you for your insightful comments. We appreciate how your input highlights potential areas of ambiguity that we can further clarify. We are open to hearing any additional thoughts or concerns you may have.

---

> > > > ### Public Comment · ~Bo_Yu9 · 2024-12-02
> > > > **Thank you for your detailed explanation.**
> > > >
> > > > Thank you for your detailed explanation; all my questions have been answered.

---

### Meta-Review · Area_Chair_UEt2 · 2024-12-19

**Metareview:**

This paper introduces an innovative approach to solving the multi-faceted Janus problem in score distillation-based text-to-3D generation methods. To tackle the bias in pose estimation towards a canonical pose, the authors establish a theoretically grounded framework that corrects the prior pose distribution, transforming it into a uniform distribution.

The paper received generally positive reviews, with scores of 6, 6, 6, and 8, leading to an average score of 6.5. All reviewers acknowledged the novelty of the work, although some expressed concerns about the limited evaluation. The authors adequately addressed most of these issues during the rebuttal. As a result, the Area Chair recommends the paper for acceptance.

**Additional Comments On Reviewer Discussion:**

The proposed method is theoretically sound and demonstrates significant novelty. Furthermore, the authors provided substantial experimental results in their rebuttal to support their claims. While most concerns were addressed, as noted by reviewer i678, there remains a lack of comparisons with other methodologies. The authors are encouraged to address this in the camera-ready version.

---

### Decision · Program_Chairs · 2025-01-22

Accept (Poster)